# One Intervention per Component is Enough: Towards Identifiability in Linear Stochastic Dynamics from Steady State

## Abstract

We study the problem of recovering the parameters of a multivariate Ornstein–Uhlenbeck (OU) process from steady-state observational and interventional data. In many applications, such as large-scale gene perturbation experiments, only stationary "snapshot" measurements are available, making standard stochastic differential equation estimation methods that rely on time-series trajectories inapplicable. We first establish an identifiability result: one intervention per strongly connected component (SCC) of the drift graph suffices to recover all OU process parameters generically up to a global scaling factor. This holds provided that the SCC condensation graph is connected with a single root and certain spectral nondegeneracy assumptions hold. We propose a recursive learning algorithm that orders SCCs topologically and, for each component, isolates its marginal dynamics and solves a linear system derived from the steady-state moment equations, leveraging parameters recovered for upstream components. Building on this theoretical foundation, we propose a regularized least-squares estimator that jointly minimizes residuals of the steady-state mean and covariance equations across observational and interventional data. Experiments on synthetic and real datasets demonstrate the effectiveness of our method in recovering parameters and predicting unseen interventions.

## 1 Introduction

Understanding the structure and parameters of dynamical systems from data is a fundamental problem in many scientific domains, including systems biology (Bansal et al., 2006; Marbach et al., 2012), neuroscience (Paninski et al., 2010), and economics (Hamilton, 2020). In many settings, such as gene regulatory networks, the underlying dynamics can be modeled by a stochastic differential equation (SDE) whose steady-state distribution encodes both the interaction structure and kinetic parameters (Gardiner, 2009; Villaverde et al., 2016). Accurately recovering these parameters enables causal representation of interactions among variables, and predictions under unseen perturbations (Pearl, 2009).

However, in experimental biology and related areas, data are often available only in the form of "snapshot" measurements (Cao et al., 2019; Schiebinger et al., 2019). The time-series trajectories (required by many SDE estimation methods (Zhang et al., 2024; Oh et al., 2024)) are expensive or infeasible to obtain at scale. In such cases, interventions such as gene knockdown (Datlinger et al., 2017) provide additional information by selectively intervening on parts of the system, potentially resolving non-identifiability issues that arise from observational data alone (Peters et al., 2017).

A substantial body of work addresses causal inference in linear stochastic systems. (Varando & Hansen, 2020; Dettling et al., 2024) focus on recovering drift structures from steady-state data using the Lyapunov equation. These approaches rely solely on observational measurements and do not leverage interventions, which can limit identifiability. More recently, Lorch et al. (2024), focused on fitting the observational/interventional distributions without providing parameter recovery guarantees. Intervention-aware models, such as in (Rohbeck et al., 2024), achieve good empirical prediction, but the identifiability results are very restricted (see the related work section for more details).

In this paper, our aim is to incorporate interventions into the learning process, establishing formal identifiability results on recovering the parameters of SDEs and developing learning algorithms grounded in these theoretical results. Our contributions are as follows:

- We establish an identifiability result for recovering the parameters of a multivariate Ornstein–Uhlenbeck (OU) process from steady-state observational and interventional data. We show that a single intervention per SCC[1] of the drift graph is sufficient to recover the parameters up to a global scaling if the SCC condensation graph is connected with a single root and certain spectral nondegeneracy assumptions (Assumptions 1,2, and 3) hold. To prove this, we first consider the case where the drift matrix is a single SCC and show that by solving the linear system of equations for the first and second moments (based on observational data and data from one intervention), we can recover all the parameters up to a global scaling (Theorem 1).

- For the general case with multiple SCCs (in Theorem 3), we provide a recursive algorithm that topologically orders the SCCs and, for each component, isolates its marginal dynamics. This reduces the problem to a sequence of single-SCC cases. For each SCC, the algorithm solves a linear system (derived from the steady-state moment equations for the observational and interventional settings) using parameters recovered for upstream components. We also provide an example that if the SCC condensation graph has more than one root, then it is impossible to learn all the parameters with a single global scaling.

- Building on these theoretical findings, we formulate a regularized least-squares optimization problem that jointly minimizes the squared residuals of the steady-state mean and covariance equations across both observational and interventional data. Empirical results on synthetic and real datasets demonstrate the effectiveness of our method in recovering parameters and predicting unseen interventions.

## 2 PROBLEM FORMULATION

**Ornstein–Uhlenbeck process**: The Ornstein–Uhlenbeck (OU) process is a continuous-time stochastic process that satisfies the stochastic differential equation (SDE):

$$d\mathbf{x} = (-\mathbf{\Lambda}\mathbf{x} + \mathbf{b})\, dt + \boldsymbol{\sigma}\, d\mathbf{W}_t, \tag{1}$$

where $\mathbf{x} \in \mathbb{R}^n$ is the state vector, $\mathbf{\Lambda} \in \mathbb{R}^{n \times n}$ is a drift matrix (assumed positive stable), $\mathbf{b} \in \mathbb{R}^n$ is a constant vector representing external input, $\boldsymbol{\sigma} \in \mathbb{R}^{n \times n}$ is the diffusion matrix, and $d\mathbf{W}_t$ is an $n$-dimensional Wiener process.

In the steady state, the OU process has:

- Mean:

$$\boldsymbol{\mu} = \mathbb{E}[\mathbf{x}_\infty] = \mathbf{\Lambda}^{-1}\mathbf{b}, \tag{2}$$

where $\mathbf{x}_\infty$ denotes a random vector distributed according to the stationary distribution of the OU process.

- Covariance:

$$\mathbf{\Lambda}\mathbf{\Sigma} + \mathbf{\Sigma}\mathbf{\Lambda}^\top = \boldsymbol{\sigma}\boldsymbol{\sigma}^\top, \tag{3}$$

where $\mathbf{\Sigma} \in \mathbb{R}^{n \times n}$ is the steady-state covariance matrix: $\mathbf{\Sigma} = \mathbb{E}[(\mathbf{x}_\infty - \boldsymbol{\mu})(\mathbf{x}_\infty - \boldsymbol{\mu})^\top]$.

We assume that the diffusion power matrix is diagonal and positive: $\mathbf{D} := \boldsymbol{\sigma}\boldsymbol{\sigma}^\top = \mathrm{diag}(d_1, \ldots, d_n) \succ 0$;

**Intervention:** We define an intervention on $i$-th coordinate of $\mathbf{x}$ as follows, where the $i$-th row of the drift matrix $\mathbf{\Lambda}$ is modified to remove influence from all other variables, while preserving its self-regulation. Formally, we define the modified drift matrix $\widetilde{\mathbf{\Lambda}}^{(i)}$ as:

$$\widetilde{\Lambda}_{kj}^{(i)} = \begin{cases} \Lambda_{kj}, & i \neq k, \\ 0, & i = k,\ j \neq i, \\ \Lambda_{ii}, & i = k,\ j = i. \end{cases}$$

---

[1]See the graph definitions in Appendix A.1.

That is, we zero out all off-diagonal entries in the $i$-th row, but retain $\Lambda_{ii}$, analogous to a knockout perturbation that isolates a variable from its regulators.

The modified dynamics are: $d\mathbf{x} = (-\widetilde{\mathbf{\Lambda}}^{(i)}\mathbf{x} + \mathbf{b})\, dt + \boldsymbol{\sigma}\, d\mathbf{W}_t$. In the intervened system, the mean is denoted by $\boldsymbol{\mu}^{(i)}$ and satisfies:

$$\boldsymbol{\mu}^{(i)} = \mathbb{E}[\mathbf{x}_\infty^{(i)}] = \left(\widetilde{\mathbf{\Lambda}}^{(i)}\right)^{-1}\mathbf{b}, \tag{4}$$

assuming $\widetilde{\mathbf{\Lambda}}^{(i)}$ is invertible.

Similarly, the steady-state covariance $\mathbf{\Sigma}^{(i)}$ satisfies the Lyapunov equation:

$$\widetilde{\mathbf{\Lambda}}^{(i)}\mathbf{\Sigma}^{(i)} + \mathbf{\Sigma}^{(i)}\left(\widetilde{\mathbf{\Lambda}}^{(i)}\right)^\top = \boldsymbol{\sigma}\boldsymbol{\sigma}^\top. \tag{5}$$

Our goal is to recover the parameters of OU process, i.e., $\mathbf{\Lambda}, \mathbf{b}$, and $\boldsymbol{\sigma}$ from the observational mean and covariance $(\boldsymbol{\mu}, \mathbf{\Sigma})$ and a collection of interventional means and covariances $\{(\boldsymbol{\mu}_i, \mathbf{\Sigma}_i)\}_{i \in \mathcal{I}}$ where $\mathcal{I}$ is the set of coordinates intervened on.

For any stationary (observational or interventional) mean and covariance, we have the following number of equations: $E := n + \frac{n(n+1)}{2} = \frac{n(n+3)}{2}$. Let $|\mathcal{I}|$ be the number of interventions (in addition to the observational one). Identifiability requires at least $(|\mathcal{I}| + 1)E \geq n^2 + 2n \implies |\mathcal{I}| \geq \left\lceil \frac{2(n+2)}{n+3} \right\rceil - 1 = 1$. Therefore, it is necessary to have at least one intervention in order to recover the parameters of the OU process. The next section establishes sufficient conditions ensuring that the parameters are identifiable, up to a global scaling, from the observational and interventional data.

## 3 IDENTIFIABILITY RESULTS

For a drift matrix $\mathbf{\Lambda}$, define the associated directed graph $G(\mathbf{\Lambda})$ by considering an edge $j \to i$ if and only if $\Lambda_{ij} \neq 0$. In the following, first we consider that $G(\mathbf{\Lambda})$ is strongly connected. Under some spectral non-degeneracy assumptions, we show that the parameters of the OU process can be recovered generically up to some global scaling by just having one intervention on any coordinate. Next, we will consider the case where $G(\mathbf{\Lambda})$ contains multiple SCCs, and show that the parameters are identifiable up to a single global scaling provided that the DAG over SCCs is connected, there is exactly one root SCC, and there is at least one intervention in each SCC.

### 3.1 SINGLE SCC

**Theorem 1.** *Consider the OU process in equation 1 with true parameters $\mathbf{\Lambda}, \mathbf{b}$, and $\boldsymbol{\sigma}$. Moreover, suppose that we have access to the observational steady-state mean and covariance $(\boldsymbol{\mu}, \mathbf{\Sigma})$ and an interventional mean and covariance $(\boldsymbol{\mu}^{(i)}, \mathbf{\Sigma}^{(i)})$ ($i$ can be any coordinate). If the graph $G(\mathbf{\Lambda})$ is strongly connected and certain spectral non-degeneracy assumptions hold (See Assumption 1 and Assumption 2 in Appendix A.5), with measure one, any other parameter triple $(\widehat{\mathbf{\Lambda}}, \widehat{\mathbf{b}}, \widehat{\mathbf{D}})$ that yields the same observational and interventional moments must satisfy:*

$$\widehat{\mathbf{\Lambda}} = c\mathbf{\Lambda}, \quad \widehat{\mathbf{b}} = c\mathbf{b}, \quad \widehat{\mathbf{D}} = c\mathbf{D},$$

*for some scalar $c > 0$.*

All the proofs of theorems (if not given in the main body) are available in the appendix. The key idea in the proof is based on forming a system of linear equations according to equation 2, equation 3, equation 4, and equation 5, showing that any possible solution for this set of equations should be a scale of the true parameters. In particular, let

$$\Theta = \begin{bmatrix} \text{vec}(\mathbf{\Lambda}) \\ \mathbf{b} \\ \mathbf{d} \end{bmatrix} \in \mathbb{R}^p,$$

where $p := n^2 + 2n$, and $\mathbf{d} := (d_1, \ldots, d_n)^\top$, where $d_k := \sigma_{kk}^2 > 0$. Moreover, the operator vec stacks the columns of its matrix argument. Therefore, $\text{vec}(\mathbf{\Lambda}) \in \mathbb{R}^{n^2}$. Now, we can write the linear equations in the following form: $\mathbf{A} \Theta = \mathbf{0}$ where $\mathbf{A} \in \mathbb{R}^{m \times p}$ (refer to Appendix A.2 for the definition of $\mathbf{A}$) and $m = n^2 + 3n$. In the proof, we show that $\mathbf{A}$ has generically a one-dimensional null space under the conditions in the statement of the theorem.

**Remark 1.** *For the analysis, we impose some spectral nondegeneracy assumptions such as the drift matrix or certain moment–based matrices have a simple spectrum (the eigenvalues being distinct). The numerical results (see Appendix C) show that these assumptions hold generically within SCCs. The statements of assumptions and their roles in the proofs are deferred to the appendix.*

## 3.2 General Case

In the general case, where the graph $G(\mathbf{\Lambda})$ has multiple SCCs, assume that the intervention set $\mathcal{I}$ contains at least one intervention targeting a variable inside each SCC. Moreover, the DAG of SCCs is connected and there is exactly one root SCC. Under these conditions and certain spectral non-degeneracy assumptions (see Assumption 2 and Assumption 3 in the appendix), we show that the parameters of the OU process can be recovered up to a global scaling. The key idea is to first recover a topological ordering over the SCCs by inspecting which means change under each intervention. The following theorem allows us to identify the set of variables in each SCC, as well as a topological ordering over the SCCs. With this structural information in hand, we then recursively identify the model parameters for each SCC by conditioning on previously resolved components, thereby reducing the multi-SCC case to a sequence of single-SCC problems.

**Theorem 2.** *Let $C$ be an SCC in $G(\mathbf{\Lambda})$ and let $i \in C$. Consider the intervention on $i$ and let $\boldsymbol{\mu}$ and $\boldsymbol{\mu}^{(i)}$ be the observational and interventional steady-state means, respectively.*

*(Non-null case). If the $i$-th row is non-null, i.e., $\exists j \neq i$ with $\Lambda_{ij} \neq 0$, then, with measure one, for the set of parameters $(\mathbf{\Lambda}, \mathbf{b})$ consistent with $G(\mathbf{\Lambda})$,*

$$\mu_k^{(i)} \neq \mu_k \quad \Longleftrightarrow \quad k \in \text{Desc}(C),$$

*where $\text{Desc}(C)$ denotes the set of nodes lying in SCCs reachable from $C$ in the DAG of SCCs (including $C$ itself).*

*(Null case). If the $i$-th row has no off-diagonals, i.e., $\Lambda_{ij} = 0$ for all $j \neq i$, then $\boldsymbol{\mu}^{(i)} = \boldsymbol{\mu}$.*

**Remark 2.** *For an intervention on $i \in C$, define $R_i := \{ j : \mu_j^{(i)} \neq \mu_j \}$. Suppose there is at least one intervention in each SCC. Based on the above theorem, intersecting and differencing the sets $R_i$ across interventions identifies the SCCs and yields a topological order for the DAG over SCCs (see Appendix A.3 for more details).*

**Theorem 3.** *Consider the OU process in equation 1 with true parameters $\mathbf{\Lambda}, \mathbf{b}, \mathbf{D}$. Suppose we have access to the observational steady-state mean and covariance $(\boldsymbol{\mu}, \mathbf{\Sigma})$, as well as to interventional means and covariances $(\boldsymbol{\mu}^{(i)}, \mathbf{\Sigma}^{(i)})$ for at least one intervention in each SCC of $G(\mathbf{\Lambda})$. If the DAG over SCCs of $G(\mathbf{\Lambda})$ is connected with a single root SCC and certain spectral non-degeneracy assumptions (see Assumption 2 and Assumption 3 in the appendix) hold, then, with measure one, any other parameter triple $(\widehat{\mathbf{\Lambda}}, \widehat{\mathbf{b}}, \widehat{\mathbf{D}})$ that yields the same observational and interventional moments must satisfy*

$$\widehat{\mathbf{\Lambda}} = c\mathbf{\Lambda}, \quad \widehat{\mathbf{b}} = c\mathbf{b}, \quad \widehat{\mathbf{D}} = c\mathbf{D},$$

*for some scalar $c > 0$.*

*Proof.* Based on Theorem 2, we can infer the SCCs and also a topological ordering over them if there is at least one intervention in each SCC. Let us denote these SCCs based on the topological ordering as $C_1, C_2, \cdots, C_K$ where $K$ is the number of components. Suppose that we already learned the parameters of the OU process in the components $C_1, C_2, \cdots, C_r$ up to some global scaling. Now, we aim for learning the parameters in $C_{r+1}$. We partition the state vector according to the SCC decomposition of $G(\mathbf{\Lambda})$:

$$\mathbf{x} = \underbrace{\mathbf{x}_P}_{C_1 \cup \cdots \cup C_r} \oplus \underbrace{\mathbf{x}_T}_{C_{r+1}} \oplus \underbrace{\mathbf{x}_F}_{C_{r+2} \cup \cdots \cup C_K},$$

where the operator $\oplus$ denotes concatenation of subvectors corresponding to disjoint index sets.

The steady-state mean and covariance matrices are partitioned accordingly:

$$\boldsymbol{\mu} = \begin{bmatrix} \boldsymbol{\mu}_P \\ \boldsymbol{\mu}_T \\ \boldsymbol{\mu}_F \end{bmatrix}, \qquad \boldsymbol{\Sigma} = \begin{bmatrix} \boldsymbol{\Sigma}_{PP} & \boldsymbol{\Sigma}_{PT} & \boldsymbol{\Sigma}_{PF} \\ \boldsymbol{\Sigma}_{TP} & \boldsymbol{\Sigma}_{TT} & \boldsymbol{\Sigma}_{TF} \\ \boldsymbol{\Sigma}_{FP} & \boldsymbol{\Sigma}_{FT} & \boldsymbol{\Sigma}_{FF} \end{bmatrix}.$$

Everything inside the $P$-block is assumed known (i.e., the blocks $\boldsymbol{\Lambda}_{PP}$, $\mathbf{b}_P$, and $\mathbf{D}_{PP}$), up to the same scaling $c$. Because $\mathbf{x}_P$ and $\mathbf{x}_T$ are jointly Gaussian, we can write:

$$\mathbf{x}_T = \mathbf{B}\,\mathbf{x}_P + \mathbf{r}, \qquad \text{where} \quad \mathbf{B} := \boldsymbol{\Sigma}_{TP}\boldsymbol{\Sigma}_{PP}^{-1}, \quad \mathbb{E}[\mathbf{r}\,\mathbf{x}_P^\top] = 0.$$

The regression matrix $\mathbf{B}$ is computable directly from the observed moments, with no dependence on model parameters. Moreover, the residual term $\mathbf{r} := \mathbf{x}_T - \mathbf{B}\mathbf{x}_P$ is a Gaussian variable with

$$\boldsymbol{\mu}_{T|P} = \boldsymbol{\mu}_T - \mathbf{B}\,\boldsymbol{\mu}_P, \qquad \boldsymbol{\Sigma}_{T|P} = \boldsymbol{\Sigma}_{TT} - \boldsymbol{\Sigma}_{TP}\boldsymbol{\Sigma}_{PP}^{-1}\boldsymbol{\Sigma}_{PT}. \tag{6}$$

Substituting the definitions of $\mathbf{B}$ and $\mathbf{r}$ into the OU dynamics yields:

$$\begin{aligned} \dot{\mathbf{r}} = -\boldsymbol{\Lambda}_{TT}\mathbf{r} &+ \left( -\boldsymbol{\Lambda}_{TP} - \boldsymbol{\Lambda}_{TT}\mathbf{B} + \mathbf{B}\boldsymbol{\Lambda}_{PP} \right)\mathbf{x}_P \\ &+ (\mathbf{b}_T - \mathbf{B}\mathbf{b}_P) + (\boldsymbol{\sigma}_T\dot{\mathbf{W}}_T - \mathbf{B}\boldsymbol{\sigma}_P\dot{\mathbf{W}}_P). \end{aligned} \tag{7}$$

Stationarity implies that the cross-covariance $\boldsymbol{\Sigma}_{rP}(t) := \mathrm{Cov}(\mathbf{r}(t), \mathbf{x}_P(t))$ is constant in time. Therefore, its derivative is zero:

$$\tfrac{d}{dt}\boldsymbol{\Sigma}_{rP}(t) = \mathbb{E}[\dot{\mathbf{r}}\,\mathbf{x}_P^\top] + \mathbb{E}[\mathbf{r}\,\dot{\mathbf{x}}_P^\top] = 0.$$

The second term vanishes as $\mathbb{E}[\mathbf{r}\,\dot{\mathbf{x}}_P^\top] = -\mathbb{E}[\mathbf{r}\,\mathbf{x}_P^\top]\,\boldsymbol{\Lambda}_{PP}^\top = \mathbf{0}$ due to the fact that $\mathbf{r}$ and $\mathbf{x}_P$ are uncorrelated. For the first term, by inserting equation 7, one obtains: $\left( -\boldsymbol{\Lambda}_{TP} - \boldsymbol{\Lambda}_{TT}\mathbf{B} + \mathbf{B}\boldsymbol{\Lambda}_{PP} \right)\boldsymbol{\Sigma}_{PP} = \mathbf{0}$. Because $\boldsymbol{\Sigma}_{PP} \succ 0$ is invertible, right-multiplication by $\boldsymbol{\Sigma}_{PP}^{-1}$ gives:

$$-\boldsymbol{\Lambda}_{TP} - \boldsymbol{\Lambda}_{TT}\mathbf{B} + \mathbf{B}\boldsymbol{\Lambda}_{PP} = \mathbf{0}. \tag{8}$$

Using equation 8 in equation 7 eliminates the $\mathbf{x}_P$ drive and leaves:

$$\dot{\mathbf{r}} = -\boldsymbol{\Lambda}_{TT}\mathbf{r} + (\mathbf{b}_T - \mathbf{B}\mathbf{b}_P) + (\boldsymbol{\sigma}_T\dot{\mathbf{W}}_T - \mathbf{B}\boldsymbol{\sigma}_P\dot{\mathbf{W}}_P), \tag{9}$$

which is an OU process acting within the block $T$.

Now, based on the first and second moments of the residual which is given in equation 6, we can identify the parameters of component $T$ and also $\boldsymbol{\Lambda}_{TP}$ up to the same scaling. In particular, we have:

$\boldsymbol{\Lambda}_{TT}$: Note that the corresponding diffusion power matrix of the residual's dynamics is: $\mathbf{D}_T + \mathbf{B}\mathbf{D}_P\mathbf{B}^\top$ and therefore, it is not necessarily diagonal. Nevertheless, the proof of Theorem 1 can be adapted to recover the $\boldsymbol{\Lambda}_{TT}$ up to the same scaling $c$ (see Appendix A.6) by solving a linear system.

$\boldsymbol{\Lambda}_{TP}$: Having $\boldsymbol{\Lambda}_{PP}$ and $\boldsymbol{\Lambda}_{TT}$ up to the scaling $c$, according to equation 8, we can also recover $\boldsymbol{\Lambda}_{TP}$ with the same scaling.

$\mathbf{b}_T$ and $\boldsymbol{D}_T$: According to equation 2, we have: $\mathbf{b}_T = \boldsymbol{\Lambda}_{TT}\boldsymbol{\mu}_T + \boldsymbol{\Lambda}_{TP}\boldsymbol{\mu}_P$. Since we recovered $\boldsymbol{\Lambda}_{TT}$ and $\boldsymbol{\Lambda}_{TP}$ with the same scaling factor, therefore, $\mathbf{b}_T$ is identifiable with the same scaling from the above equation. Regarding $\boldsymbol{\sigma}_T$ (or diffusion power matrix $\mathbf{D}_T$), from the Lyapunov equation for the covariance matrix of residual (in other words, $\boldsymbol{\Sigma}_{T|P}$), we have:

$$\boldsymbol{\Lambda}_{TT}\boldsymbol{\Sigma}_{T|P} + \boldsymbol{\Sigma}_{T|P}\boldsymbol{\Lambda}_{TT}^\top = \mathbf{D}_T + \mathbf{B}\,\mathbf{D}_P\,\mathbf{B}^\top.$$

Therefore, $\mathbf{D}_T$ can be learned with the same scaling and this completes the proof. $\qquad\square$

**Remark 3.** *The two structural assumptions in Theorem 3 are necessary. If the DAG over SCCs is disconnected, then each disconnected part of the DAG can only be identified up to its own scaling factor. Moreover, if there are multiple root SCCs, the parameters of the OU process cannot, in general, be recovered up to a single global scaling. An example illustrating this case is provided in Appendix A.7.*

Building on the above, we design a recursive algorithm that proceeds in two stages (the pseudo-code is given in Algorithm 1):

- In the first stage, we use the changes in interventional means to identify the SCCs of the drift graph $G(\boldsymbol{\Lambda})$, along with a topological ordering over these components. This structural information is inferred using Theorem 2.

- In the second stage, we iterate over the SCCs in topological order. For each component $T$, we treat the union of its ancestral SCCs as $P$, and condition on $\mathbf{x}_P$ to isolate the marginal dynamics of $\mathbf{x}_T$. Using the conditional moments $(\boldsymbol{\mu}_{T|P}, \boldsymbol{\Sigma}_{T|P})$, we first recover $\boldsymbol{\Lambda}_{TT}$ up to a scaling. Then, leveraging the structure of the OU dynamics, we recover $\boldsymbol{\Lambda}_{TP}$, $\boldsymbol{b}_T$, and $\boldsymbol{D}_T$ up to the same scaling. The procedure continues recursively until all components have been identified.

---

**Algorithm 1** Recursive OU Learning with Interventions

---

1: **Input:** Observational and interventional means and covariances $(\boldsymbol{\mu}, \boldsymbol{\Sigma}), \{(\boldsymbol{\mu}^{(i)}, \boldsymbol{\Sigma}^{(i)})\}_{i \in \mathcal{I}}$
2: **Phase 1: Learn SCCs**
3: Identify SCCs, $C_1, C_2, \ldots, C_K$, and a topological ordering over them from mean changes using Theorem 2
4: **Phase 2: Recover parameters per SCC**
5: **for** each component $T = C_j$ in topological order **do**
6:     Let $P := C_1 \cup \cdots \cup C_{j-1}$ be the union of ancestor SCCs
7:     Compute conditional mean $\boldsymbol{\mu}_{T|P}$ and covariance $\boldsymbol{\Sigma}_{T|P}$ according to equation 6
8:     Recover $\boldsymbol{\Lambda}_{TT}$ using the interventional moments with the same scaling in $P$
9:     Recover $\boldsymbol{\Lambda}_{TP}$, $\boldsymbol{b}_T$, $\boldsymbol{D}_T$ up to the same scaling using cross-covariances and stationary conditions
10: **end for**
11: **Output:** Drift matrix $\boldsymbol{\Lambda}$, input vector $\boldsymbol{b}$, and diffusion matrix $\boldsymbol{D}$ (up to a global scaling)

---

**Remark 4.** *Rather than directly running Algorithm 1, we can alternatively construct a linear system of equations based on the true first and second moments of the observational and interventional settings, similar to the approach used in the proof of Theorem 1. If the linear system has a one-dimensional null space, then all parameters can be identified up to a scaling factor. Theorem 2 establishes that the null space is one-dimensional if the conditions in the statement of the theorem are satisfied.*

## 4 LEARNING ALGORITHM

In finite-sample settings, plugging in empirical moments generally yields full-column-rank systems, which admit only the trivial solution of zero vector. Nevertheless, the identifiability result in Theorem 1 motivates replacing true moments with empirical ones and relaxing the equations into a least-squares objective. Specifically, in the observational case, the mean vector $\boldsymbol{\mu}$ and covariance matrix $\boldsymbol{\Sigma}$ satisfy the linear system in equation 2 and equation 3, respectively. Moreover, for each intervention $i \in \mathcal{I}$, the mean vector $\boldsymbol{\mu}^{(i)}$ and $\boldsymbol{\Sigma}^{(i)}$ satisfy equations in equation 4 and equation 5, respectively.

For the set of free parameters $\Theta = (\text{vec}(\boldsymbol{\Lambda}), \mathbf{b}, \mathbf{d})$, where $\mathbf{d}$ is the diagonal of matrix $\mathbf{D}$, we define the following least-square objective:

$$
\mathcal{L}(\Theta) = \alpha_O \left( \|\boldsymbol{\Lambda}\,\hat{\boldsymbol{\mu}} - \mathbf{b}\|_2^2 + \left\|\boldsymbol{\Lambda}\,\hat{\boldsymbol{\Sigma}} + \hat{\boldsymbol{\Sigma}}\,\boldsymbol{\Lambda}^\top - \mathbf{D}\right\|_F^2 \right)
$$
$$
+ \alpha_I \sum_{i \in \mathcal{I}} \left( \left\|\widetilde{\boldsymbol{\Lambda}}^{(i)}\,\hat{\boldsymbol{\mu}}^{(i)} - \mathbf{b}\right\|_2^2 + \left\|\widetilde{\boldsymbol{\Lambda}}^{(i)}\,\hat{\boldsymbol{\Sigma}}^{(i)} + \hat{\boldsymbol{\Sigma}}^{(i)}(\widetilde{\boldsymbol{\Lambda}}^{(i)})^\top - \mathbf{D}\right\|_F^2 \right), \tag{10}
$$

where $\alpha_O, \alpha_I > 0$ are weighting coefficients; we often give a larger weight to observational terms since observational samples are often more abundant and their estimates are more accurate. Moreover, $\hat{\boldsymbol{\mu}}, \hat{\boldsymbol{\Sigma}}, \hat{\boldsymbol{\mu}}^{(i)}, \hat{\boldsymbol{\Sigma}}^{(i)}, i \in \mathcal{I}$ are the unbiased estimates of first and second moments in the observational and interventional settings and $\|\cdot\|_F$ is the Frobenius norm. To promote sparsity in the drift graph, we add an $\ell_1$ penalty on the off-diagonal elements of $\boldsymbol{\Lambda}$: $\mathcal{R}(\boldsymbol{\Lambda}) = \gamma \sum_{i \neq j} |\Lambda_{ij}|$, where $\gamma > 0$ controls the sparsity level. Therefore, the optimization problem becomes

$$
\min_\Theta \mathcal{L}(\Theta) + \mathcal{R}(\boldsymbol{\Lambda}), \tag{11}
$$

subject to stability constraints (such as $\Lambda_{kk} > 0$ and $d_k > 0$ for all $k$).

## 5 RELATED WORK

Herein, we mainly review methods that perform inference from stationary distributions of dynamical systems, rather than from full trajectories.

**Identifiability and learning from steady state in SDE models.** A line of work on graphical continuous Lyapunov models treats the steady-state covariance as the solution of a continuous Lyapunov equation. In this setting, Varando & Hansen (2020) proposed an $l_1$-regularized estimator to recover sparse drift structure from observational snapshots. Dettling et al. (2023) subsequently analyzed identifiability in this framework, proving that when only observational covariances are available and the diffusion matrix is known, the drift is globally identifiable from the covariance if and only if the drift graph is simple, meaning it contains no directed two-cycles. While these results provide valuable insights, they are limited to observational data, cannot recover models with two-cycles in the drift, and do not handle unknown diffusion. In contrast, our work incorporates interventional data, allows unknown diagonal diffusion, and shows that a single intervention in each SCC (under some conditions on the DAG of SCCs) suffices for recovery up to a global scaling. More recently, Dettling et al. (2024) proposed a Lasso-based estimator for recovering the drift structure of linear SDEs from stationary observational covariances and analyzed its identifiability properties. However, their framework does not incorporate interventions. Very recently, Zweig et al. (2025) studied identifiability of linear SDEs under mean-shift interventions, assuming a low-rank drift matrix. In contrast, we allow general drift matrices and base identifiability on the SCC structure of the drift graph rather than on a low rank assumption.

Other approaches aim to learn stationary SDE models without focusing on identifiability. Lorch et al. (2024) proposed a kernel deviation-from-stationarity objective that measures how far a candidate SDE's stationary distribution deviates from the empirical distribution. Their framework can accommodate cycles and generalizes well to unseen interventions, but its goal is density fitting in reproducing kernel Hilbert spaces rather than moment-based parameter identification. Our work differs by deriving explicit graphical conditions under which the OU parameters are recoverable from first and second moments.

Recent empirical work has also combined steady-state dynamics with interventions. Rohbeck et al. (2024) introduced Bicycle, a model in which interventions alter a subset of parameters. Bicycle achieves good performance in both structure recovery and prediction under out-of-distribution interventions in single-cell datasets. However, it is designed primarily as a predictive model and provides identifiability guarantees only when interventions are performed on all coordinates except one. In contrast, we gave identifiability results under substantially weaker requirements, i.e., one intervention per SCC.

Boege et al. (2025) have studied the conditional independence relations implied by sparsity in the drift of stationary multivariate diffusions. Their results link graph structure to conditional independencies in the stationary state distribution. Nonetheless, this line of work does not address the recovery of drift or diffusion parameters, nor how interventions could enable identifiability.

Finally, there exists related work for deterministic linear ODEs rather than stochastic OU models. For example, Wang et al. (2024) analyzed the identifiability of linear ODE systems with hidden confounders from time series or discretely sampled trajectories and derived conditions under which latent confounding can be resolved.

**Gene perturbation prediction from steady state.** In systems biology, several methods aim to predict the effects of genetic perturbations directly from stationary data. For instance, Sethuraman et al. (2023) proposed NODAGS-Flow, which learns nonlinear cyclic causal structures from interventional steady-state gene expression by fitting residual normalizing flows, producing predictions and plausible graphs. While providing good performance in practice, NODAGS-Flow is a likelihood-based method without formal identifiability guarantees on recovering the parameters of the underlying system.

Other works focus on high-performing predictive models. For instance, Roohani et al. (2022) proposed GEAR, which combines graph neural networks with prior biological network information to predict transcriptional responses to single or multiple gene perturbations, showing improved generalization to unseen combinations. Yu et al. (2025) proposed PerturbNet, which uses conditional invertible flows to model the distributional effects of unseen perturbations. PerturBench (Wu et al.)

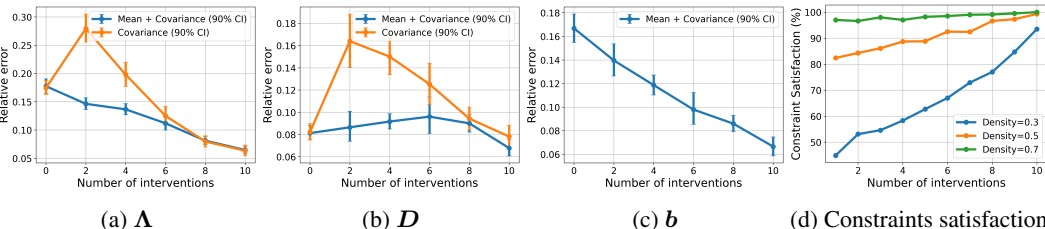

(a) $\mathbf{\Lambda}$        (b) $\boldsymbol{D}$        (c) $\boldsymbol{b}$        (d) Constraints satisfaction

Figure 1: (a–c) Relative errors of estimated parameters of the OU process versus the number of interventions, and (d) percentage of instances satisfying graphical conditions in Theorem 3.

provides a unified benchmark suite for single-cell perturbation modeling, facilitating comparison between predictive models.

## 6 EXPERIMENTS

**Synthetic Data:** We generated synthetic datasets by simulating steady-state observations from a stable linear stochastic system. The drift matrix $\mathbf{\Lambda} \in \mathbb{R}^{n \times n}$ was initialized as a zero matrix, then each off-diagonal entry was set to a Gaussian random value with probability $\rho$ and left at zero otherwise, where $\rho$ is the desired density level. For each row, the diagonal entry was set to be larger than the sum of the absolute values of off-diagonal entries in that row by at least a positive margin, ensuring stability. The diffusion power $\mathbf{D}$ was diagonal with strictly positive entries sampled uniformly from $[d_{\min} = 0.2, d_{\max} = 0.4]$, and the bias vector $\mathbf{b}$ was drawn uniformly from $[b_{\min} = 0.2, b_{\max} = 1.5]$. The observational steady-state mean and covariance were then computed by solving the corresponding linear and Lyapunov equations for the given $\mathbf{\Lambda}$, $\mathbf{b}$, and $\mathbf{D}$. Interventions were simulated by zeroing all off-diagonal entries in a selected row of $\mathbf{\Lambda}$ while keeping the diagonal entry unchanged and leaving $\mathbf{b}$ unchanged. For the observational setting and each intervention, samples were drawn from the corresponding multivariate normal distribution. All the implementations are available in the following link: https://anonymous.4open.science/r/OU_ID-278C. More details of experiments are given in Appendix B.

In Figure 1 (a-c), we report the estimation error of $\mathbf{\Lambda}$, $\mathbf{D}$, and $\mathbf{b}$ as a function of the number of interventions with $n = 10$. To evaluate recovery up to scaling, we compute the scaling factor $c = \langle \hat{\mathbf{A}}, \mathbf{A} \rangle / \langle \mathbf{A}, \mathbf{A} \rangle$ for each parameter matrix/vector $\mathbf{A} \in \{\mathbf{\Lambda}, \mathbf{D}, \mathbf{b}\}$, where $\langle \mathbf{X}, \mathbf{Y} \rangle = \mathrm{tr}(\mathbf{X}^\top \mathbf{Y})$ for matrices and $\langle \mathbf{x}, \mathbf{y} \rangle = \mathbf{x}^\top \mathbf{y}$ for vectors. The relative error is then measured as $\|\hat{\mathbf{A}} - c\mathbf{A}\| / \|\mathbf{A}\|$. We apply this procedure separately to $\mathbf{\Lambda}$, $\mathbf{D}$, and $\mathbf{b}$. The results with 90% confidence intervals are given in blue curves with the legend "Mean and Covariance (90% CI)". As shown, for $\mathbf{\Lambda}$ and $\mathbf{b}$, the relative error decreases as the number of interventions increases. For $\mathbf{D}$, we observe a small non-monotone effect: the error is initially low with no interventions, increases slightly for a few interventions, and then decreases again. One possible explanation is that $\mathbf{D}$ is already estimated accurately from observational data, so incorporating a small number of interventional data (which may contain fewer samples than the observational data) can worsen its estimation. As the number of interventions increases, however, the estimation of $\mathbf{\Lambda}$ improves, which in turn reduces the error in $\mathbf{D}$.

We also report results when the loss includes only the residuals of the Lyapunov equations (i.e., using covariances but not means) with the legend "Covariance (90% CI)," similar to (Dettling et al., 2024; Rohbeck et al., 2024). This variant performs noticeably worse, highlighting that the mean terms are essential (please note that there is no curve for $\mathbf{b}$ in this case as there is no term for estimating it); mean estimates are typically much more accurate than covariance estimates, and incorporating them substantially improves recovery.

In Figure 1d, we depict the percentage of instances of $\mathbf{\Lambda}$ satisfying the graphical conditions in Theorem 3 as a function of the number of interventions for different graph densities ($\rho$). For sparse graphs ($\rho = 0.3$), less than 50% of instances satisfy the conditions with a single intervention, but the percentage increases steadily as more interventions are added. In contrast, denser graphs ($\rho = 0.5, 0.7$) already satisfy the graphical conditions at a high rate with only a few interventions.

Table 1: Evaluation of DES and PDS on the three Perturb-seq datasets. The interventional mean is shaded to indicate oracle access to interventional data.

| Method | Co-Culture | | Control | | IFN-$\gamma$ | |
| --- | --- | --- | --- | --- | --- | --- |
| | DES | PDS | DES | PDS | DES | PDS |
| Observational mean | 0.33 | 0.57 | 0.33 | 0.57 | 0.35 | 0.67 |
| Ours ("Covariance") | 0.51 | 0.43 | **0.46** | 0.43 | 0.42 | 0.43 |
| Ours ("Mean + Covariance") | 0.43 | **0.67** | 0.33 | 0.63 | **0.47** | **0.70** |
| Interventional mean | **0.52** | 0.57 | 0.42 | **0.67** | 0.36 | **0.70** |

**Real Data:** We assess our method on real-world data, leveraging three published single-cell perturbation screen datasets (Frangieh et al., 2021). Since the true causal graph is unknown in this setting, our evaluation focuses on generalization to unseen perturbations. Specifically, we consider a Perturb-seq dataset containing targeted CRISPR knock-out perturbations of 249 target genes in tumor-infiltrating lymphocytes (TILs) of melanoma patients. The perturbations were performed under three conditions, which we treat as separate datasets: a baseline culture of TILs in a neutral medium ("Control"), a culture of TILs with interferon-$\gamma$ added ("IFN-$\gamma$"), and a co-culture of TILs with patient-derived melanoma cells ("Co-Culture").

Following the setup in (Sethuraman et al., 2023), we restrict our analysis to the same subset of 61 genes and adopt their reported training/test split: 90% of interventions are used for training and the remaining 10% are held out for evaluation, with analyses performed separately for each dataset. Our goal is to predict the interventional mean $\mu$ for unseen perturbations, using the estimated parameters and the steady-state equation for the mean. Note that global scaling is not an issue here, as it cancels out for predicting $\mu$.

For evaluation, we do not rely on Mean Absolute Error (MAE), as even the observational mean achieves a very close performance to the one using the interventional mean on the held-out set. Instead, following the recommendation in the Virtual Cell Challenge[2], we report the Differential Expression Score (DES) and the Perturbation Differential Score (PDS), where higher values indicate better performance. DES measures agreement in identifying differentially expressed genes, while PDS measures a model's ability to distinguish between perturbations by ranking predictions according to their similarity to the true perturbational effect, regardless of their effect size. In Table 1, we compare our full method ("Mean + Covariance") against the "Covariance" only variant (similar to the approaches in (Dettling et al., 2024; Rohbeck et al., 2024)), and against the one using the interventional mean of the held-out set (which is not available to our method). As shown in Table 1, our method achieves comparable and in some cases even higher scores than the oracle baseline using the interventional mean, despite not having access to held-out interventional data.

## 7 CONCLUSIONS AND FUTURE WORK

We studied recovery of multivariate OU parameters from steady-state observational and interventional data. Our main theoretical contribution shows that one intervention per SCC of the drift graph suffices for generic recovery of $(\Lambda, \mathbf{b}, \mathbf{D})$ up to a single global scaling when the DAG over SCCs is connected with a unique root. The single-SCC case (Theorem 1) yields a rank-1 null space for the stacked moment equations, and the multi-SCC result (Theorem 3) follows via a constructive, recursive decomposition that leverages mean shifts (Theorem 2) to infer SCCs and a topological order. Building on these guarantees, we proposed a regularized least-squares estimator that fits the steady-state mean and covariance equations jointly across observational and interventional data, and we observed accurate parameter recovery in synthetic datasets. Our guarantees rely on some spectral nondegeneracy assumptions, such as the drift matrix having a simple spectrum. Numerical results suggest these hold generically for strongly connected drift graphs. A key avenue for future work is a formal genericity proof. Another direction is to understand how diffusion assumptions (e.g., not diagonal diffusion matrices) change the boundary between identifiable and non-identifiable regimes.

---

[2]https://virtualcellchallenge.org/evaluation#scoring

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

# A APPENDIX

## A.1 GRAPH DEFINITIONS

**Strongly connected components.** For a directed graph $G$, a *strongly connected component (SCC)* is a maximal subset of nodes $C$ such that for every pair $u, v \in C$ there exists a directed path from $u$ to $v$ and a directed path from $v$ to $u$. The SCCs of $G$ form a partition of its nodes.

**Condensation graph.** Given the SCCs $C_1, \ldots, C_K$ of $G$, the *condensation graph* is a directed graph whose nodes correspond to the SCCs and which contains a directed edge $C_i \to C_j$ whenever there exists an edge in $G$ from any node in $C_i$ to any node in $C_j$. By construction, the condensation graph is always a directed acyclic graph (DAG).

**Topological order over SCCs.** A *topological order* of the SCCs is any ordering of the nodes of the condensation graph such that all directed edges point from earlier to later components. Equivalently, $C_i$ may appear before $C_j$ in the order if and only if there is no directed path from $C_j$ to $C_i$ in the condensation graph.

## A.2 FORMING THE LINEAR SYSTEM

Throughout, $\mathbf{I}_n$ denotes the $n \times n$ identity, $\otimes$ is the Kronecker product, and $\mathbf{e}_k$ is the $k$-th canonical basis vector in $\mathbb{R}^n$.

We define the following selector matrices:

- Introduce the matrix $\mathbf{P} \in \mathbb{R}^{n^2 \times n}$ such that $\text{vec}(\text{diag}(\mathbf{d})) = \mathbf{P}\,\mathbf{d}$, where the $k$-th column of $\mathbf{P}$ is given by $\mathbf{P}_{:,k} = \text{vec}(\mathbf{E}_{kk})$, and $\mathbf{E}_{kk}$ denotes the $n \times n$ matrix with a one in entry $(k, k)$ and zeros elsewhere.

- For a fixed row index $i$, let $\mathbf{E}_i := \mathbf{I}_n \otimes \mathbf{e}_i^\top \in \mathbb{R}^{n \times n^2}$. Therefore, $\mathbf{E}_i \text{vec}(\mathbf{\Lambda}) = \left[\Lambda_{i1}, \ldots, \Lambda_{in}\right]^\top$ extracts the entire $i$-th row of $\mathbf{\Lambda}$.

- $\mathbf{S}_n \in \mathbb{R}^{\frac{n(n+1)}{2} \times n^2}$ is the upper-triangular elimination matrix.

- Let $\mathbf{C} \in \mathbb{R}^{n^2 \times n^2}$ denote the commutation matrix, so that $\text{vec}(\mathbf{X}^\top) = \mathbf{C}\,\text{vec}(\mathbf{X})$ for any $X \in \mathbb{R}^{n \times n}$.

For the equation of observational mean, we define:

$$\mathbf{M}_0 := \left[-\left(\boldsymbol{\mu}^\top \otimes \mathbf{I}_n\right) \,\middle|\, \mathbf{I}_n \,\middle|\, \mathbf{0}\right] \in \mathbb{R}^{n \times p},$$

where $p = n^2 + 2n$.

For the equation of the interventional mean (on coordinate $i$), let $\mathbf{J}_i := \mathbf{I}_n - \mathbf{e}_i \mathbf{e}_i^\top$. We define:

$$\mathbf{M}_1 := \left[-\left((\boldsymbol{\mu}^{(i)})^\top \otimes \mathbf{I}_n\right) + \mathbf{e}_i(\boldsymbol{\mu}^{(i)})^\top \mathbf{J}_i \mathbf{E}_i \,\middle|\, \mathbf{I}_n \,\middle|\, \mathbf{0}\right] \in \mathbb{R}^{n \times p}.$$

For the observational covariance, vectorising $\mathbf{\Lambda\Sigma} + \mathbf{\Sigma\Lambda}^\top - \text{diag}(\mathbf{d}) = 0$ and selecting the upper-triangular part yields

$$\mathbf{K}_0 = \left[\mathbf{S}_n\left(\mathbf{\Sigma} \otimes \mathbf{I}_n + (\mathbf{I}_n \otimes \mathbf{\Sigma})\,\mathbf{C}\right) \,\middle|\, \mathbf{0} \,\middle|\, -\mathbf{S}_n\mathbf{P}\right] \in \mathbb{R}^{\frac{n(n+1)}{2} \times p}.$$

For the interventional covariance (on coordinate $i$), we define:

$$\mathbf{K}_1 = \left[\mathbf{S}_n\left(\mathbf{\Sigma}^{(i)} \otimes \mathbf{I}_n + (\mathbf{I}_n \otimes \mathbf{\Sigma}^{(i)})\,\mathbf{C} - \left[(\mathbf{I}_n \otimes \mathbf{e}_i) + (\mathbf{e}_i \otimes \mathbf{I}_n)\right]\mathbf{\Sigma}^{(i)}\mathbf{J}_i\mathbf{E}_i\right) \,\middle|\, \mathbf{0} \,\middle|\, -\mathbf{S}_n\mathbf{P}\right] \in \mathbb{R}^{\frac{n(n+1)}{2} \times p}.$$

Assembling all the equations:

$$\mathbf{A} := \begin{bmatrix} \mathbf{M}_0 \\ \mathbf{M}_1 \\ \mathbf{K}_0 \\ \mathbf{K}_1 \end{bmatrix} \in \mathbb{R}^{m \times p}, \tag{12}$$

where $m = n + n + \frac{n(n+1)}{2} + \frac{n(n+1)}{2} = n^2 + 3n$.

### A.3   PROOF OF THEOREM 2

At steady state, $\boldsymbol{\mu} = \boldsymbol{\Lambda}^{-1}\mathbf{b}$ and $\boldsymbol{\mu}^{(i)} = (\widetilde{\boldsymbol{\Lambda}}^{(i)})^{-1}\mathbf{b}$. Let $\mathbf{E} := \widetilde{\boldsymbol{\Lambda}}^{(i)} - \boldsymbol{\Lambda}$; only the $i$-th row of $\mathbf{E}$ is nonzero, with $E_{ij} = -\Lambda_{ij}$ for $j \neq i$. By the following equation,

$$\Delta\boldsymbol{\mu} := \boldsymbol{\mu} - \boldsymbol{\mu}^{(i)} = \boldsymbol{\Lambda}^{-1}\mathbf{E}\,(\widetilde{\boldsymbol{\Lambda}}^{(i)})^{-1}\mathbf{b} = \boldsymbol{\Lambda}^{-1}\mathbf{E}\,\boldsymbol{\mu}^{(i)}. \tag{13}$$

Since only row $i$ of $\mathbf{E}$ is nonzero, $\mathbf{E}\mathbf{v} = s(\mathbf{v})\,\mathbf{e}_i$ for any vector $\mathbf{v}$, where

$$s(\mathbf{v}) := -\sum_{j \neq i} \Lambda_{ij}\,v_j.$$

Applying this to $\mathbf{v} = \boldsymbol{\mu}^{(i)}$ in equation 13 gives

$$\Delta\boldsymbol{\mu} = s(\boldsymbol{\mu}^{(i)})\,\boldsymbol{\Lambda}^{-1}\mathbf{e}_i. \tag{14}$$

Permute coordinates by some permutation matrix that orders the SCCs topologically. Let $\mathbf{L}$ be the block lower–triangular drift matrix after topologically ordering the SCCs,

$$\mathbf{L} = \begin{bmatrix} \mathbf{L}^{(1)} & 0 & \cdots & 0 \\ \mathbf{L}^{(2,1)} & \mathbf{L}^{(2)} & \ddots & \vdots \\ \vdots & \ddots & \ddots & 0 \\ \mathbf{L}^{(K,1)} & \cdots & \mathbf{L}^{(K,K-1)} & \mathbf{L}^{(K)} \end{bmatrix},$$

with each diagonal block $\mathbf{L}^{(u)}$ invertible. Fix a block $r$ and a coordinate $i$ in block $r$. Solving $\mathbf{L}\,\mathbf{y} = \mathbf{e}_i$ by forward substitution yields: (i) $\mathbf{y}^{(t)} = \mathbf{0}$ for all blocks $t$ that are *not* descendants of $r$; (ii) $\mathbf{y}^{(r)} \neq \mathbf{0}$; and (iii) for any descendant block $t$ of $r$, $\mathbf{y}^{(t)}$ is a sum of terms that are multilinear polynomials in the inter-block entries along directed paths from $r$ to $t$ and in $\mathbf{y}^{(r)}$. In particular, we show that if $t$ is a descendant of $r$ in the DAG over SCCs and $j$ is a coordinate in block $t$, then the scalar map $\Theta \longmapsto (\mathbf{L}(\Theta)^{-1}\mathbf{e}_i)_j$ is not identically zero on the admissible parameter set $\mathcal{P}$ of matrices consistent with the block sparsity (with invertible diagonal blocks). In other words, $(\mathbf{L}^{-1}\mathbf{e}_i)_j \neq 0$ for generic $\Theta \in \mathcal{P}$.

Write $\mathbf{L} = \mathbf{D} + \mathbf{N}$, where $\mathbf{D} = \mathrm{blockdiag}(\mathbf{L}^{(1)}, \ldots, \mathbf{L}^{(K)})$ and $\mathbf{N}$ is strictly block–lower (i.e., $\mathbf{N}^{(t,\ell)} = \mathbf{L}^{(t,\ell)}$ for $t > \ell$ and 0 otherwise). Then

$$\mathbf{L}^{-1} = (\mathbf{D} + \mathbf{N})^{-1} = (\mathbf{I} + \mathbf{D}^{-1}\mathbf{N})^{-1}\mathbf{D}^{-1} = \sum_{m=0}^{K-1} (-1)^m (\mathbf{D}^{-1}\mathbf{N})^m \mathbf{D}^{-1},$$

because $(\mathbf{D}^{-1}\mathbf{N})^m = 0$ for all $m \geq K$ (as $\mathbf{D}^{-1}\mathbf{N}$ is strict block–lower nilpotent). The $(t, r)$ block of $\mathbf{L}^{-1}$ is therefore a finite sum of terms of the form

$$\mathbf{D}_t^{-1}\mathbf{N}_{t,s_{k-1}}\mathbf{D}_{s_{k-1}}^{-1}\cdots\mathbf{N}_{s_1,r}\mathbf{D}_r^{-1},$$

indexed by block paths $r = s_0 \to s_1 \to \cdots \to s_k = t$ in the DAG over SCCs. Hence each entry of $(\mathbf{L}^{-1})_{t,r}$ is a rational function of the free parameters.

To prove the function $\Theta \mapsto (\mathbf{L}(\Theta)^{-1}\mathbf{e}_i)_j$ is not identically zero, it suffices to exhibit one admissible $\Theta^\star \in \mathcal{P}$ with $(\mathbf{L}(\Theta^\star)^{-1}\mathbf{e}_i)_j \neq 0$.[3] Fix a particular block path $r = s_0 \to s_1 \to \cdots \to s_k = t$. Construct a one-parameter family $\mathbf{L}(\varepsilon) \in \mathcal{P}$ as follows, choosing any constants $\alpha_u \neq 0$.

---

[3] On $\{\Theta : \det\mathbf{L}(\Theta) \neq 0\}$, the adjugate formula gives $\mathbf{L}(\Theta)^{-1} = \mathrm{adj}(\mathbf{L}(\Theta))/\det\mathbf{L}(\Theta)$, so $(\mathbf{L}(\Theta)^{-1})_{j,i} = C_{j,i}(\Theta)/\det\mathbf{L}(\Theta)$ where $C_{j,i}$ is the $(j, i)$-cofactor (a polynomial in the entries of $\mathbf{L}(\Theta)$). Hence each entry of $\mathbf{L}(\Theta)^{-1}$ is a rational function of $\Theta$.

*(i) Diagonal blocks.* For each block $u$, set

$$\mathbf{L}^{(u)}(\varepsilon) = \alpha_u \mathbf{I} + \varepsilon \mathbf{B}^{(u)},$$

where $\mathbf{B}^{(u)}$ is supported exactly on the allowed in-block sparsity and has all its allowed entries equal to 1. For $\varepsilon$ small, each $\mathbf{L}^{(u)}(\varepsilon)$ is invertible and $\mathbf{D}_u(\varepsilon)^{-1} = \mathbf{L}^{(u)}(\varepsilon)^{-1} = \alpha_u^{-1}\mathbf{I} + O(\varepsilon)$.

*(ii) Inter-block blocks along the chosen path.* For each edge $s_{m-1} \to s_m$, pick one allowed position in $\mathbf{L}^{(s_m, s_{m-1})}$ and set that entry to 1; set all other allowed entries in that block to $\varepsilon$.

*(iii) All other inter-block blocks.* Set all their allowed entries to $\varepsilon$.

By construction, every allowed entry of $\mathbf{L}(\varepsilon)$ is nonzero for $\varepsilon > 0$, and $\mathbf{L}(\varepsilon)$ remains block-lower with invertible diagonal blocks, hence $\mathbf{L}(\varepsilon) \in \mathcal{P}$.

Let $\mathbf{y}(\varepsilon) := \mathbf{L}(\varepsilon)^{-1}\mathbf{e}_i$ and consider the scalar

$$f_j(\varepsilon) := \big(\mathbf{y}(\varepsilon)\big)_j = \big(\mathbf{L}(\varepsilon)^{-1}\mathbf{e}_i\big)_j, \qquad j \in \text{block } t.$$

Expanding $\mathbf{L}(\varepsilon)^{-1}$ with the finite series above, the contribution of the *chosen* block path contains the term

$$T_{\text{path}}(\varepsilon) := \mathbf{e}_j^\top \mathbf{D}_t(\varepsilon)^{-1} \mathbf{N}_{t,s_{k-1}}(\varepsilon) \mathbf{D}_{s_{k-1}}(\varepsilon)^{-1} \cdots \mathbf{N}_{s_1,r}(\varepsilon) \mathbf{D}_r(\varepsilon)^{-1}\mathbf{e}_i.$$

By the parameter choice, each $\mathbf{N}_{s_m,s_{m-1}}(\varepsilon)$ has a designated entry equal to 1 (all its other allowed entries are $\varepsilon$), and $\mathbf{D}_u(\varepsilon)^{-1} = \alpha_u^{-1}\mathbf{I} + O(\varepsilon)$. Therefore the *constant term* of $T_{\text{path}}(\varepsilon)$ equals

$$c_0 = \pm \alpha_t^{-1} \alpha_{s_{k-1}}^{-1} \cdots \alpha_r^{-1} \neq 0,$$

obtained by multiplying the designated 1's on the inter-blocks and the $\alpha_u^{-1}$ from the diagonals. Every other contribution to $f_j(\varepsilon)$ (from other block paths or from non-designated entries, which are $\varepsilon$) carries at least one factor of $\varepsilon$ and is thus $O(\varepsilon)$ in that scalar entry. Hence

$$f_j(\varepsilon) = c_0 + \varepsilon\, c_1(\varepsilon), \qquad c_0 \neq 0,$$

for some analytic $c_1$ near $\varepsilon = 0$. For sufficiently small $\varepsilon > 0$, $f_j(\varepsilon) \neq 0$.

We have thus exhibited admissible parameters with $(\mathbf{L}^{-1}\mathbf{e}_i)_j \neq 0$. Consequently, the analytic function $\Theta \mapsto \big(\mathbf{L}(\Theta)^{-1}\mathbf{e}_i\big)_j$ is not identically zero on $\mathcal{P}$, so its zero set is contained in a real analytic subset (in particular, of Lebesgue measure zero). Equivalently, for generic $\Theta \in \mathcal{P}$, $\big(\mathbf{L}^{-1}\mathbf{e}_i\big)_j \neq 0$.

If for some $j \neq i$, we have $\Lambda_{ij} \neq 0$, then

$$s\big(\boldsymbol{\mu}^{(i)}\big) = -\sum_{j \neq i} \Lambda_{ij}\, \mu_j^{(i)}$$

is an analytic, not-identically-zero function of $(\boldsymbol{\Lambda}, \mathbf{b})$ (as $\mathbf{b}$ is all non-zero vector, $\boldsymbol{\mu}^{(i)}$ is generically non-zero). Hence $s(\boldsymbol{\mu}^{(i)}) \neq 0$ generically. Combining with equation 14, the support of $\Delta\boldsymbol{\mu}$ matches that of $\boldsymbol{\Lambda}^{-1}\mathbf{e}_i$ whenever $s(\boldsymbol{\mu}^{(i)}) \neq 0$. Thus, if for some $j \neq i$, $\Lambda_{ij} \neq 0$, generically,

$$\mu_j^{(i)} \neq \mu_j \iff j \in C \cup \text{Desc}(C).$$

If $\Lambda_{ij} = 0$ for all $j \neq i$ (i.e., $i$ is single node SCC), then $s(\boldsymbol{\mu}^{(i)})$ is zero and then we can detect it from $\boldsymbol{\mu}^{(i)} = \boldsymbol{\mu}$.

## A.4 Recovering SCCs and a topological order over SCCs from mean changes

Assume at least one intervention is performed in every SCC. For each intervention on $i$, set

$$R_i := \{\, j : \mu_j^{(i)} \neq \mu_j \,\}.$$

By Theorem 2, generically,

$$R_i = \text{Desc}(C_i) \quad \text{if the intervention on } i \text{ is non-null}, \qquad R_i = \emptyset \quad \text{if it is null},$$

where $C_i$ is the SCC containing $i$ and $\text{Desc}(C)$ is the set of nodes lying in SCCs reachable from $C$ in the SCC–DAG (including $C$ itself).

**Procedure.**

1. *Singleton sources (null interventions).* If $R_i = \emptyset$, declare $\{i\}$ a singleton SCC with no incoming edges (a source). Do *not* merge different $i$ with $R_i = \emptyset$.

2. *Non-null SCCs.* For the remaining $i$, group by equality of sets: $i \sim i'$ iff $R_i = R_{i'}$. Each equivalence class is exactly one SCC; write $R_C$ for the common set of a class $C$.

3. *Edges among non-null SCCs.* There is an edge from $C$ to $C'$ if $R_C \supsetneq R_{C'}$ and no $D$ satisfying $R_C \supsetneq R_D \supsetneq R_{C'}$.

4. *Topological order.* Output all singleton sources from step 1 first (in any arbitrary order), then the non-null SCCs according to the recovered edges among non-null SCCs. This yields a valid topological ordering.

## A.5  Proof of Theorem 1

Consider the following two admissible triples (i.e., satisfying linear system in equation 12).

- True underlying triple: $(\mathbf{\Lambda}_\star, \mathbf{b}_\star, \mathbf{D}_\star)$
- Alternative admissible triple: $(\mathbf{\Lambda}', \mathbf{b}', \mathbf{D}')$

Define the differences:

$$\Delta\mathbf{\Theta} := \begin{bmatrix} \mathrm{vec}(\Delta\mathbf{\Lambda}) \\ \Delta\mathbf{b} \\ \Delta\mathbf{d} \end{bmatrix}, \qquad \Delta\mathbf{\Lambda} := \mathbf{\Lambda}' - \mathbf{\Lambda}_\star, \quad \Delta\mathbf{b} := \mathbf{b}' - \mathbf{b}_\star, \quad \Delta\mathbf{D} := \mathbf{D}' - \mathbf{D}_\star,$$

where $\Delta\mathbf{d}$ is the diagonal of $\Delta\mathbf{D}$.

Since $\mathbf{A}\mathbf{\Theta}_\star = 0$ and $\mathbf{A}\mathbf{\Theta}' = 0$, it follows that $\mathbf{A}\Delta\mathbf{\Theta} = 0$. Our goal is to show that $\Delta\mathbf{\Theta}$ is a *scalar multiple* of $\mathbf{\Theta}_\star$, i.e.,

$$\mathbf{\Lambda}' = c\mathbf{\Lambda}_\star, \quad \mathbf{b}' = c\mathbf{b}_\star, \quad \mathbf{D}' = c\mathbf{D}_\star,$$

for some scalar $c$.

Fix the intervened row index $i$ throughout the proof. The observational blocks $\mathbf{M}_0, \mathbf{K}_0$ yield:

$$\Delta\mathbf{\Lambda}\,\boldsymbol{\mu} = \Delta\mathbf{b}, \qquad \Delta\mathbf{\Lambda}\,\mathbf{\Sigma} + \mathbf{\Sigma}\,\Delta\mathbf{\Lambda}^\top = \Delta\mathbf{D}, \tag{15}$$

where $\boldsymbol{\mu}, \mathbf{\Sigma}$ are the true observational steady-state mean and covariance.

Since the intervention zeros out row $i$ off-diagonals, we have:

$$\widetilde{\mathbf{\Lambda}}_\star^{(i)} = \mathbf{J}_i \odot \mathbf{\Lambda}_\star, \quad \widetilde{\mathbf{\Lambda}}'^{(i)} = \mathbf{J}_i \odot \mathbf{\Lambda}', \quad \text{where } \mathbf{J}_i := \mathbf{1}\mathbf{1}^\top - \mathbf{e}_i\mathbf{1}^\top + \mathbf{e}_i\mathbf{e}_i^\top,$$

where $\odot$ is the Hadamard product. Therefore, the corresponding equations for $\mathbf{M}_1, \mathbf{K}_1$ become:

$$(\mathbf{J}_i \odot \Delta\mathbf{\Lambda})\,\boldsymbol{\mu}^{(i)} = \Delta\mathbf{b}, \qquad (\mathbf{J}_i \odot \Delta\mathbf{\Lambda})\,\mathbf{\Sigma}^{(i)} + \mathbf{\Sigma}^{(i)}\,(\Delta\mathbf{\Lambda}^\top \odot \mathbf{J}_i^\top) = \Delta\mathbf{D} \tag{16}$$

Taking the $i$-th row of the first equation:

$$(\Delta\mathbf{\Lambda})_{ii}\,\mu_i^{(i)} = \Delta b_i. \tag{17}$$

Let us define

$$c := \frac{b_i'}{b_{\star i}} \quad \text{(assumed well-defined for } b_{\star i} \neq 0\text{)}.$$

Then, we define:

$$\widehat{\mathbf{\Lambda}} := \mathbf{\Lambda}' - c\mathbf{\Lambda}_\star, \quad \widehat{\mathbf{b}} := \mathbf{b}' - c\mathbf{b}_\star, \quad \widehat{\mathbf{D}} := \mathbf{D}' - c\mathbf{D}_\star.$$

Observe that:

- $\widehat{b}_i = 0$.
- The new triple $\widehat{\mathbf{\Theta}} = (\widehat{\mathbf{\Lambda}}, \widehat{\mathbf{b}}, \widehat{\mathbf{D}})$ satisfies $\mathbf{A}\widehat{\mathbf{\Theta}} = 0$.

From now on, for simplifying notations, we drop the hats. From equation 17 and $b_i = 0$, we get $\Lambda_{ii} = 0$. Moreover, according to $\mathbf{K}_1$, for the entry $(i, i)$, we have: $2\Sigma_{ii}^{(i)}\Lambda_{ii} = D_{ii}$. Therefore, $D_{ii} = 0$.

As $D_{ii} = 0$ and $\Lambda_{ii} = 0$, then from equation $\mathbf{K}_1$, we have:

$$\widetilde{\boldsymbol{\Lambda}}^{(i)} = \begin{bmatrix} 0 & 0 \\ \boldsymbol{\Lambda}_{-i,i} & \boldsymbol{\Lambda}_{-i,-i} \end{bmatrix}, \qquad \boldsymbol{\Sigma}^{(i)} = \begin{bmatrix} \boldsymbol{\Sigma}_{ii}^{(i)} & \boldsymbol{\Sigma}_{i,-i}^{(i)} \\ \boldsymbol{\Sigma}_{-i,i}^{(i)} & \boldsymbol{\Sigma}_{-i,-i}^{(i)} \end{bmatrix}.$$

Therefore,

$$\widetilde{\boldsymbol{\Lambda}}^{(i)}\,\boldsymbol{\Sigma}^{(i)} + \boldsymbol{\Sigma}^{(i)}\,(\widetilde{\boldsymbol{\Lambda}}^{(i)})^\top =$$

$$\begin{bmatrix} 0 & \boldsymbol{\Sigma}_{ii}^{(i)}\,\boldsymbol{\Lambda}_{-i,i}^\top + \boldsymbol{\Sigma}_{i,-i}^{(i)}\,\boldsymbol{\Lambda}_{-i,-i}^\top \\ \boldsymbol{\Lambda}_{-i,i}\,\boldsymbol{\Sigma}_{ii}^{(i)} + \boldsymbol{\Lambda}_{-i,-i}\,\boldsymbol{\Sigma}_{-i,i}^{(i)} & \boldsymbol{\Lambda}_{-i,i}\,\boldsymbol{\Sigma}_{i,-i}^{(i)} + \boldsymbol{\Lambda}_{-i,-i}\,\boldsymbol{\Sigma}_{-i,-i}^{(i)} + \boldsymbol{\Sigma}_{-i,i}^{(i)}\,\boldsymbol{\Lambda}_{-i,i}^\top + \boldsymbol{\Sigma}_{-i,-i}^{(i)}\,\boldsymbol{\Lambda}_{-i,-i}^\top \end{bmatrix}.$$

We know that the off-diagonal entries of the above matrix should be zero. Therefore, we have: $\boldsymbol{\Lambda}_{-i,i} = -(\boldsymbol{\Lambda}_{-i,-i}\,\boldsymbol{\Sigma}_{-i,i}^{(i)})/\boldsymbol{\Sigma}_{ii}^{(i)}$. Hence, the bottom right block becomes: $\boldsymbol{\Lambda}_{-i,-i}\mathbf{Z} + \mathbf{Z}\boldsymbol{\Lambda}_{-i,-i}^\top$ where $\mathbf{Z} = \boldsymbol{\Sigma}_{-i,-i}^{(i)} - (\boldsymbol{\Sigma}_{-i,i}^{(i)}\boldsymbol{\Sigma}_{i,-i}^{(i)})/\boldsymbol{\Sigma}_{ii}^{(i)}$ (which is positive definite as it is Schur complement of $\boldsymbol{\Sigma}_{ii}^{(i)}$ and $\boldsymbol{\Sigma}^{(i)}$ is positive definite and $\boldsymbol{\Sigma}_{ii}^{(i)} \neq 0$).

Now, from the two Lyapunov equations:

$$\boldsymbol{\Lambda}\boldsymbol{\Sigma} + \boldsymbol{\Sigma}\boldsymbol{\Lambda}^\top = \mathbf{D}, \tag{18}$$

$$\widetilde{\boldsymbol{\Lambda}}^{(i)}\boldsymbol{\Sigma}^{(i)} + \boldsymbol{\Sigma}^{(i)}\widetilde{\boldsymbol{\Lambda}}^{(i)\top} = \mathbf{D}, \tag{19}$$

where

$$\widetilde{\boldsymbol{\Lambda}}^{(i)} = \boldsymbol{\Lambda} - \mathbf{e}_i\mathbf{w}^\top, \quad \text{with} \quad \mathbf{w} := \boldsymbol{\Lambda}_{i,-i}^\top.$$

Subtracting the second equation from the first and defining

$$\boldsymbol{\Gamma} := \boldsymbol{\Sigma} - \boldsymbol{\Sigma}^{(i)},$$

we obtain:

$$\boldsymbol{\Lambda}\boldsymbol{\Gamma} + \boldsymbol{\Gamma}\boldsymbol{\Lambda}^\top + \mathbf{e}_i\mathbf{w}^\top\boldsymbol{\Sigma}^{(i)} + \boldsymbol{\Sigma}^{(i)}\mathbf{w}\mathbf{e}_i^\top = 0. \tag{20}$$

Insert $\boldsymbol{\Lambda} = \widetilde{\boldsymbol{\Lambda}}^{(i)} + \mathbf{e}_i\mathbf{w}^\top$ into the above equation:

$$\widetilde{\boldsymbol{\Lambda}}^{(i)}\boldsymbol{\Gamma} + \boldsymbol{\Gamma}\widetilde{\boldsymbol{\Lambda}}^{(i)\top} + \mathbf{e}_i\mathbf{w}^\top\boldsymbol{\Gamma} + \boldsymbol{\Gamma}\mathbf{w}\mathbf{e}_i^\top + \mathbf{e}_i\mathbf{w}^\top\boldsymbol{\Sigma}^{(i)} + \boldsymbol{\Sigma}^{(i)}\mathbf{w}\mathbf{e}_i^\top = 0.$$

Note that the four terms involving $\mathbf{e}_i$ (all on the right-hand side) have nonzero entries only in row $i$ and column $i$. Therefore, in the $(-i, -i)$ sub-block, they vanish, leaving:

$$\left(\widetilde{\boldsymbol{\Lambda}}^{(i)}\boldsymbol{\Gamma} + \boldsymbol{\Gamma}\widetilde{\boldsymbol{\Lambda}}^{(i)\top}\right)_{-i,-i} = 0. \tag{21}$$

Use the $\{i\} \cup -i$ block partitioning:

$$\widetilde{\boldsymbol{\Lambda}}^{(i)} = \begin{bmatrix} 0 & 0 \\ \boldsymbol{\Lambda}_{-i,i} & \boldsymbol{\Lambda}_{-i,-i} \end{bmatrix}, \quad \boldsymbol{\Gamma} = \begin{bmatrix} \boldsymbol{\Gamma}_{ii} & \boldsymbol{\Gamma}_{i,-i} \\ \boldsymbol{\Gamma}_{-i,i} & \boldsymbol{\Gamma}_{-i,-i} \end{bmatrix}.$$

Multiplying:

$$\widetilde{\boldsymbol{\Lambda}}^{(i)}\boldsymbol{\Gamma} = \begin{bmatrix} 0 & \boldsymbol{\Lambda}_{ii}\boldsymbol{\Gamma}_{i,-i} \\ \boldsymbol{\Lambda}_{-i,-i}\boldsymbol{\Gamma}_{-i,i} & \boldsymbol{\Lambda}_{-i,i}\boldsymbol{\Gamma}_{i,-i} + \boldsymbol{\Lambda}_{-i,-i}\boldsymbol{\Gamma}_{-i,-i} \end{bmatrix}.$$

Similarly,

$$\boldsymbol{\Gamma}\widetilde{\boldsymbol{\Lambda}}^{(i)\top} = \begin{bmatrix} 0 & \boldsymbol{\Gamma}_{i,-i}\boldsymbol{\Lambda}_{-i,-i}^\top \\ \boldsymbol{\Gamma}_{-i,i}\boldsymbol{\Lambda}_{ii} & \boldsymbol{\Gamma}_{-i,i}\boldsymbol{\Lambda}_{-i,i}^\top + \boldsymbol{\Gamma}_{-i,-i}\boldsymbol{\Lambda}_{-i,-i}^\top \end{bmatrix}.$$

Add the $(-i, -i)$ blocks from both results and apply equation 21:

$$\mathbf{\Lambda}_{-i,i}\mathbf{\Gamma}_{i,-i} + \mathbf{\Lambda}_{-i,-i}\mathbf{\Gamma}_{-i,-i} + \mathbf{\Gamma}_{-i,i}\mathbf{\Lambda}_{-i,i}^\top + \mathbf{\Gamma}_{-i,-i}\mathbf{\Lambda}_{-i,-i}^\top = 0. \tag{22}$$

By $\mathbf{\Lambda}_{-i,i} = -(\mathbf{\Lambda}_{-i,-i}\,\mathbf{\Sigma}_{-i,i}^{(i)})/\mathbf{\Sigma}_{ii}^{(i)}$, we get:

$$\mathbf{\Lambda}_{-i,-i}\,\mathbf{\Xi} + \mathbf{\Xi}^\top\mathbf{\Lambda}_{-i,-i}^\top = 0, \tag{23}$$

where

$$\mathbf{\Xi} := \mathbf{\Gamma}_{-i,-i} - \frac{\mathbf{\Sigma}_{-i,i}^{(i)}\,\mathbf{\Gamma}_{i,-i}}{\mathbf{\Sigma}_{ii}^{(i)}}. \tag{24}$$

We consider the two equations derived above involving the parameters $\mathbf{\Lambda}_{-i,-i}$ and $\mathbf{D}_{-i,-i}$:

$$\mathbf{\Lambda}_{-i,-i}\mathbf{Z} + \mathbf{Z}\mathbf{\Lambda}_{-i,-i}^\top = \mathbf{D}_{-i,-i}, \tag{25}$$

$$\mathbf{\Lambda}_{-i,-i}\mathbf{\Xi} + \mathbf{\Xi}^\top\mathbf{\Lambda}_{-i,-i}^\top = \mathbf{0}, \tag{26}$$

where $\mathbf{Z} \in \mathbb{R}^{(n-1)\times(n-1)}$ is symmetric and positive definite, and $\mathbf{\Xi} \in \mathbb{R}^{(n-1)\times(n-1)}$.

Define $\mathbf{A} := \mathbf{\Xi}^{-1}\mathbf{Z}$, and $\mathbf{S} := \mathbf{\Lambda}_{-i,-i}\mathbf{\Xi}$. Then: 1- $\mathbf{S}$ is skew-symmetric; 2- $\mathbf{SA} - \mathbf{A}^\top\mathbf{S} = \mathbf{D}_{-i,-i}$ is diagonal, hence $\mathrm{offdiag}(\mathbf{SA} - \mathbf{A}^\top\mathbf{S}) = 0$.

**Assumption 1.** *The matrix* $\mathbf{A} := \mathbf{\Xi}^{-1}\mathbf{Z}$*, has a simple (pairwise-distinct) spectrum.*

Considering above assumption, take the Schur decomposition $\mathbf{A} = \mathbf{Q}\mathbf{T}\mathbf{Q}^\top$ with $\mathbf{Q}$ orthogonal and $\mathbf{T}$ upper triangular with pairwise-distinct diagonal entries $(\zeta_1, \ldots, \zeta_{n-1})$. Set $\widetilde{\mathbf{S}} = \mathbf{Q}^\top\mathbf{S}\mathbf{Q}$; then $\widetilde{\mathbf{S}}^\top = -\widetilde{\mathbf{S}}$ and

$$\mathrm{offdiag}(\mathbf{SA} - \mathbf{A}^\top\mathbf{S}) = 0 \iff \forall r \neq s: \; (\widetilde{\mathbf{S}}\mathbf{T} - \mathbf{T}^\top\widetilde{\mathbf{S}})_{rs} = 0.$$

For $r < s$,

$$(\widetilde{\mathbf{S}}\mathbf{T} - \mathbf{T}^\top\widetilde{\mathbf{S}})_{rs} = (\zeta_s - \zeta_r)\widetilde{\mathbf{S}}_{rs} + \sum_{j=r+1}^{s-1}(\widetilde{\mathbf{S}}_{rj}\mathbf{T}_{js} - \mathbf{T}_{jr}\widetilde{\mathbf{S}}_{js}). \tag{27}$$

In the following, we show by induction on the gap $s - r$, $\widetilde{\mathbf{S}}_{rs} = 0$ for all $r < s$. Hence $\widetilde{\mathbf{S}} = \mathbf{0}$ by skewness, and therefore $\mathbf{S} = \mathbf{0}$.

*Base case $g = 1$.* Here $s = r + 1$. The sum in equation 27 is empty because there is no index $j$ strictly between $r$ and $s$. Thus,

$$0 = (\zeta_{r+1} - \zeta_r)\widetilde{\mathbf{S}}_{r,r+1}.$$

Since the eigenvalues are distinct, $\zeta_{r+1} - \zeta_r \neq 0$, so $\widetilde{\mathbf{S}}_{r,r+1} = 0$.

*Inductive step.* Assume that for some $g > 1$ we already know $\widetilde{\mathbf{S}}_{ab} = 0$ for all pairs $a < b$ with $b - a < g$. Now fix $r < s$ with $s - r = g$. In the sum in equation 27, every term involves either $\widetilde{\mathbf{S}}_{rj}$ with $j - r < g$ or $\widetilde{\mathbf{S}}_{js}$ with $s - j < g$. By the induction hypothesis, all these terms vanish. Therefore the entire sum in equation 27 is zero, leaving $0 = (\zeta_s - \zeta_r)\widetilde{\mathbf{S}}_{rs} = 0$. Again $\zeta_s \neq \zeta_r$, so $\widetilde{\mathbf{S}}_{rs} = 0$.

Now, since $\mathbf{S} = 0$, according to $\mathbf{S} := \mathbf{\Lambda}_{-i,-i}\mathbf{\Xi}$ and the fact that $\mathbf{\Xi}$ is generically invertible (see Lemma 1), $\mathbf{\Lambda}_{-i,-i}$ is also zero. Now, from equation 25, $\mathbf{D}_{-i,-i}$ become zero and therefore the whole $\mathbf{D}$ is zero. Moreover, from $\mathbf{K}_0$, we have: $\mathbf{\Lambda} = 0$ and due to $\mathbf{M}_0$, we have: $\mathbf{b} = \mathbf{0}$ and the proof of theorem is complete.

**Assumption 2.** *Consider right/left eigenbases* $\{\mathbf{r}_\ell\}_{\ell=1}^n$, $\{\boldsymbol{\rho}_k\}_{k=1}^n$ *with* $\mathbf{\Lambda}\mathbf{r}_\ell = \lambda_\ell\mathbf{r}_\ell$, $\boldsymbol{\rho}_k^\top\mathbf{\Lambda} = \lambda_k\boldsymbol{\rho}_k^\top$, $\boldsymbol{\rho}_k^\top\mathbf{r}_\ell = \delta_{k\ell}$.

1. *We assume that* $\mathbf{\Lambda}$ *has simple spectrum, i.e.,* $\lambda_k \neq \lambda_l$ *for any* $k \neq l$.

2. *Let* $\mathbf{R} := [\mathbf{r}_1 \cdots \mathbf{r}_n]$, $\mathbf{P} := [\boldsymbol{\rho}_1 \cdots \boldsymbol{\rho}_n]$ *so that* $\mathbf{P}^\top \mathbf{R} = \mathbf{I}$. *Let* $\mathbf{W}_\Lambda \in \mathbb{R}^{n \times n}$ *be:*

$$(\mathbf{W}_\Lambda)_{k\ell} := \frac{(\boldsymbol{\rho}_k^\top \mathbf{e}_i)(\mathbf{w}^\top \boldsymbol{\Sigma}^{(i)} \mathbf{r}_\ell)}{\lambda_k + \lambda_\ell}. \tag{28}$$

*We assume that* $-1 \notin \sigma(\mathbf{W}_\Lambda^{-T} \mathbf{W}_\Lambda)$. *Therefore, whenever* $\mathbf{W}_\Lambda$ *is invertible,* $\det(\mathbf{W}_\Lambda + \mathbf{W}_\Lambda^\top) = \det(\mathbf{W}_\Lambda^\top(\mathbf{I} + \mathbf{W}_\Lambda^{-T} \mathbf{W}_\Lambda)) \neq 0$.

**Lemma 1.** *Under Assumption 2, the matrix* $\boldsymbol{\Xi}$ *is generically invertible.*

*Proof.* Let $\mathbf{U}$ solve the Sylvester equation

$$\boldsymbol{\Lambda} \mathbf{U} + \mathbf{U} \boldsymbol{\Lambda}^\top = -\mathbf{e}_i (\mathbf{w}^\top \boldsymbol{\Sigma}^{(i)}). \tag{29}$$

Since $\lambda_k + \lambda_\ell \neq 0$ for all $k, \ell$ (positive stability), the Sylvester operator is invertible and equation 29 has a unique solution. Transposing equation 29 and adding yields

$$\boldsymbol{\Lambda}(\mathbf{U} + \mathbf{U}^\top) + (\mathbf{U} + \mathbf{U}^\top)\boldsymbol{\Lambda}^\top = -\mathbf{e}_i \mathbf{w}^\top \boldsymbol{\Sigma}^{(i)} - \boldsymbol{\Sigma}^{(i)} \mathbf{w} \mathbf{e}_i^\top.$$

Comparing with equation 20 and invoking uniqueness, we obtain

$$\boldsymbol{\Gamma} = \mathbf{U} + \mathbf{U}^\top. \tag{30}$$

Expand $\mathbf{U} = \sum_{k,\ell} u_{k\ell} \, \mathbf{r}_k \boldsymbol{\rho}_\ell^\top$. Multiplying equation 29 with $\boldsymbol{\rho}_k^\top(\cdot)$ from the left and $\mathbf{r}_\ell$ from the right gives

$$(\lambda_k + \lambda_\ell) \, u_{k\ell} = -(\boldsymbol{\rho}_k^\top \mathbf{e}_i)(\mathbf{w}^\top \boldsymbol{\Sigma}^{(i)} \mathbf{r}_\ell).$$

Hence $u_{k\ell} = -(\mathbf{W}_\Lambda)_{k\ell}$ with $(\mathbf{W}_\Lambda)_{k\ell}$ defined in equation 28. In matrix form, we can write:

$$\mathbf{U} = -\mathbf{R} \, \mathbf{W}_\Lambda \, \mathbf{P}^\top. \tag{31}$$

Using equation 30,

$$\boldsymbol{\Gamma} = \mathbf{U} + \mathbf{U}^\top = -\mathbf{R} \, \mathbf{W}_\Lambda \, \mathbf{P}^\top - \mathbf{P} \, \mathbf{W}_\Lambda^\top \mathbf{R}^\top. \tag{32}$$

Premultiplying by $\mathbf{P}^\top$ and postmultiplying by $\mathbf{R}$,

$$\mathbf{P}^\top \boldsymbol{\Gamma} \mathbf{R} = -(\mathbf{W}_\Lambda + \mathbf{W}_\Lambda^\top). \tag{33}$$

For strongly connected $G(\boldsymbol{\Lambda})$ with nonzero self-loops, by PBH controllability, $\boldsymbol{\rho}_k^\top \mathbf{e}_i \neq 0$ for all $k$; by PBH observability, $\mathbf{w}^\top \boldsymbol{\Sigma}^{(i)} \mathbf{r}_\ell \neq 0$ for all $\ell$. Therefore

$$\mathbf{W}_\Lambda = \underbrace{\mathrm{diag}(\boldsymbol{\rho}_k^\top \mathbf{e}_i)}_{=: \, \mathbf{D}_L \text{ invertible}} \cdot \underbrace{\left[\frac{1}{\lambda_k + \lambda_\ell}\right]_{k,\ell}}_{=: \, \mathbf{C} \text{ (Cauchy, invertible)}} \cdot \underbrace{\mathrm{diag}(\mathbf{w}^\top \boldsymbol{\Sigma}^{(i)} \mathbf{r}_\ell)}_{=: \, \mathbf{D}_R \text{ invertible}},$$

$\mathbf{W}_\Lambda = \mathbf{D}_L \, \mathbf{C} \, \mathbf{D}_R$ is invertible as $\mathbf{D}_L, \mathbf{D}_R$ are invertible by PBH, and the Cauchy matrix $\mathbf{C}$ is invertible since $\lambda_k + \lambda_\ell \neq 0$ (positive stability) and $\boldsymbol{\Lambda}$ has simple spectrum (Assumption 2 (1)). Based on Assumption 2 (2), this implies $\boldsymbol{\Gamma}$ is invertible.

Let $J = \{1, \ldots, n\} \setminus \{i\}$ and $\mathbf{P}_J = \mathbf{I} - \mathbf{e}_i \mathbf{e}_i^\top$. The block system

$$\begin{bmatrix} \boldsymbol{\Sigma}_{ii}^{(i)} & \boldsymbol{\Gamma}_{i,J} \\ \boldsymbol{\Sigma}_{J,i}^{(i)} & \boldsymbol{\Gamma}_{J,J} \end{bmatrix} \begin{pmatrix} x_0 \\ \mathbf{x} \end{pmatrix} = \mathbf{0}$$

is equivalent to

$$\boldsymbol{\Gamma} \mathbf{z}_J = -\boldsymbol{\Sigma}^{(i)} \mathbf{e}_i x_0, \qquad \mathbf{z}_J := \mathbf{P}_J[x_0; \mathbf{x}], \quad (\mathbf{z}_J)_i = 0, \tag{34}$$

by stacking the $i$ and $J$ rows. Since $\boldsymbol{\Gamma}$ is invertible, equation 34 gives $\mathbf{z}_J = -\boldsymbol{\Gamma}^{-1} \boldsymbol{\Sigma}^{(i)} \mathbf{e}_i x_0$. Taking the $i$-th coordinate and using $(\mathbf{z}_J)_i = 0$,

$$0 = \mathbf{e}_i^\top \mathbf{z}_J = -(\mathbf{e}_i^\top \boldsymbol{\Gamma}^{-1} \boldsymbol{\Sigma}^{(i)} \mathbf{e}_i) x_0.$$

Assuming $\mathbf{e}_i^\top \boldsymbol{\Gamma}^{-1} \boldsymbol{\Sigma}^{(i)} \mathbf{e}_i$ is generically non-zero, hence $x_0 = 0$, and thus $\mathbf{z}_J = \mathbf{0}$ and $\mathbf{x} = \mathbf{0}$. The kernel is trivial, so by the Schur identity, the determinant of the matrix in the block system (equal to $\boldsymbol{\Sigma}_{ii}^{(i)} \det \boldsymbol{\Xi}$) with $\boldsymbol{\Sigma}_{ii}^{(i)} \neq 0$ is non-zero. Therefore, we have $\det \boldsymbol{\Xi} \neq 0$. $\qquad\square$

### A.6    PROOF OF THEOREM 3

Let $\mathbf{T}$ be the target block. Suppose the parameters of block $\mathbf{P}$ satisfy $\mathbf{b}_{\mathbf{P}} = c\,\bar{\mathbf{b}}_{\mathbf{P}}$ and $\mathbf{D}_{\mathbf{P}} = c\,\bar{\mathbf{D}}_{\mathbf{P}}$ for a scalar $c > 0$ and known representatives $\bar{\mathbf{b}}_{\mathbf{P}}, \bar{\mathbf{D}}_{\mathbf{P}}$. Define the residual covariances and cross terms

$$\mathbf{Z} := \boldsymbol{\Sigma}_{\mathbf{T}|\mathbf{P}}, \quad \mathbf{Z}^{(i)} := \boldsymbol{\Sigma}_{\mathbf{T}|\mathbf{P}}^{(i)}, \quad \mathbf{B} := \boldsymbol{\Sigma}_{\mathbf{TP}}\boldsymbol{\Sigma}_{\mathbf{PP}}^{-1}, \quad \mathbf{B}^{(i)} := \boldsymbol{\Sigma}_{\mathbf{TP}}^{(i)}\big(\boldsymbol{\Sigma}_{\mathbf{PP}}^{(i)}\big)^{-1},$$

and set

$$\mathbf{v} := \mathbf{B}\,\bar{\mathbf{b}}_{\mathbf{P}}, \quad \mathbf{v}^{(i)} := \mathbf{B}^{(i)}\bar{\mathbf{b}}_{\mathbf{P}}, \qquad \mathbf{Q} := \mathbf{B}\,\bar{\mathbf{D}}_{\mathbf{P}}\,\mathbf{B}^{\top}, \quad \mathbf{Q}^{(i)} := \mathbf{B}^{(i)}\bar{\mathbf{D}}_{\mathbf{P}}\,\mathbf{B}^{(i)\top}.$$

Under the intervention on coordinate $i$, the $i$-th row of $\mathbf{B}^{(i)}$ is zero, hence $(\mathbf{Q}^{(i)})_{i,-i} = 0$. Partition the drift on $\mathbf{T}$ as

$$\boldsymbol{\Lambda}_{\mathbf{TT}} = \begin{bmatrix} \lambda_{ii} & \boldsymbol{\lambda}_{i,-i} \\ \boldsymbol{\lambda}_{-i,i} & \boldsymbol{\Lambda}_{-i,-i} \end{bmatrix}, \qquad \widetilde{\boldsymbol{\Lambda}}_{\mathbf{TT}}^{(i)} = \begin{bmatrix} \lambda_{ii} & 0 \\ \boldsymbol{\lambda}_{-i,i} & \boldsymbol{\Lambda}_{-i,-i} \end{bmatrix},$$

with unknown diagonal $\mathbf{D}_{\mathbf{T}}$ and vector $\mathbf{b}_{\mathbf{T}}$.

On $\mathbf{T}$, the stationary equations are

$$\boldsymbol{\Lambda}_{\mathbf{TT}}\mathbf{Z} + \mathbf{Z}\boldsymbol{\Lambda}_{\mathbf{TT}}^{\top} = \mathbf{D}_{\mathbf{T}} + c\,\mathbf{Q}, \qquad \boldsymbol{\Lambda}_{\mathbf{TT}}\boldsymbol{\mu}_{\mathbf{T}|\mathbf{P}} = \mathbf{b}_{\mathbf{T}} - c\,\mathbf{v},$$

and, under intervention,

$$\widetilde{\boldsymbol{\Lambda}}_{\mathbf{TT}}^{(i)}\mathbf{Z}^{(i)} + \mathbf{Z}^{(i)}\widetilde{\boldsymbol{\Lambda}}_{\mathbf{TT}}^{(i)\top} = \mathbf{D}_{\mathbf{T}} + c\,\mathbf{Q}^{(i)}, \qquad \widetilde{\boldsymbol{\Lambda}}_{\mathbf{TT}}^{(i)}\boldsymbol{\mu}_{\mathbf{T}|\mathbf{P}}^{(i)} = \mathbf{b}_{\mathbf{T}} - c\,\mathbf{v}^{(i)}. \tag{35}$$

Subtracting gives the difference relations

$$\boldsymbol{\Lambda}_{\mathbf{TT}}\mathbf{Z} + \mathbf{Z}\boldsymbol{\Lambda}_{\mathbf{TT}}^{\top} - \big(\widetilde{\boldsymbol{\Lambda}}_{\mathbf{TT}}^{(i)}\mathbf{Z}^{(i)} + \mathbf{Z}^{(i)}\widetilde{\boldsymbol{\Lambda}}_{\mathbf{TT}}^{(i)\top}\big) = c\,(\mathbf{Q} - \mathbf{Q}^{(i)}),$$

$$\boldsymbol{\Lambda}_{\mathbf{TT}}\boldsymbol{\mu}_{\mathbf{T}|\mathbf{P}} - \widetilde{\boldsymbol{\Lambda}}_{\mathbf{TT}}^{(i)}\boldsymbol{\mu}_{\mathbf{T}|\mathbf{P}}^{(i)} = c\,(\mathbf{v}^{(i)} - \mathbf{v}).$$

**Interventional $(i, -i)$ block.**   Taking the $(i, -i)$ block of the interventional Lyapunov equation and using that $\mathbf{D}_{\mathbf{T}}$ is diagonal and $(\mathbf{Q}^{(i)})_{i,-i} = 0$ yields

$$\lambda_{ii}\,\mathbf{Z}_{-i,i}^{(i)} + \boldsymbol{\lambda}_{-i,i}\,Z_{ii}^{(i)} + \boldsymbol{\Lambda}_{-i,-i}\,\mathbf{Z}_{-i,i}^{(i)} = 0.$$

Since $Z_{ii}^{(i)} > 0$, write $\boldsymbol{u} := \mathbf{Z}_{-i,i}^{(i)}/Z_{ii}^{(i)}$ to obtain

$$\boldsymbol{\lambda}_{-i,i} + (\boldsymbol{\Lambda}_{-i,-i} + \lambda_{ii}\mathbf{I})\,\boldsymbol{u} = 0\,. \tag{36}$$

**The $\boldsymbol{\Xi}$-equation on $(-i, -i)$.**   Let $\boldsymbol{\Gamma} := \mathbf{Z} - \mathbf{Z}^{(i)}$ and define the data-only matrix

$$\boldsymbol{\Xi} := \boldsymbol{\Gamma}_{-i,-i} - \boldsymbol{u}\,\boldsymbol{\Gamma}_{i,-i}.$$

From the $(-i, -i)$ block of the difference Lyapunov equation and the identity above,

$$\boldsymbol{\Lambda}_{-i,-i}\,\boldsymbol{\Xi} + \boldsymbol{\Xi}^{\top}\boldsymbol{\Lambda}_{-i,-i}^{\top} = c\,(\mathbf{Q} - \mathbf{Q}^{(i)})_{-i,-i} + \lambda_{ii}\left(\boldsymbol{u}\,\boldsymbol{\Gamma}_{i,-i} + \boldsymbol{\Gamma}_{-i,i}\,\boldsymbol{u}^{\top}\right). \tag{37}$$

From equation 35 and equation 36, we derive

$$\mathrm{offdiag}\Big(\boldsymbol{\Lambda}_{-i,-i}\,\mathbf{S}^{(i)} + \mathbf{S}^{(i)}\,\boldsymbol{\Lambda}_{-i,-i}^{\top}\Big) - \lambda_{ii}\,\mathrm{offdiag}\Big(\boldsymbol{u}\,\mathbf{Z}_{i,-i}^{(i)} + \mathbf{Z}_{-i,i}^{(i)}\,\boldsymbol{u}^{\top}\Big) = \mathrm{offdiag}\big(c\,\mathbf{Q}_{-i,-i}^{(i)}\big), \tag{38}$$

with $\boldsymbol{u} := \mathbf{Z}_{-i,i}^{(i)}/Z_{ii}^{(i)}$ and $\mathbf{S}^{(i)} := \mathbf{Z}_{-i,-i}^{(i)} - \boldsymbol{u}\,\mathbf{Z}_{i,-i}^{(i)} \succ 0$. Define the linear maps

$$\mathcal{A}_S(\mathbf{X}) := \mathrm{offdiag}(\mathbf{X}\mathbf{S}^{(i)} + \mathbf{S}^{(i)}\mathbf{X}^{\top}), \qquad \mathcal{A}_{\boldsymbol{\Xi}}(\mathbf{X}) := \mathbf{X}\boldsymbol{\Xi} + \boldsymbol{\Xi}^{\top}\mathbf{X}^{\top}.$$

Equations equation 37 and equation 38 form a stacked linear system in $\boldsymbol{\Lambda}_{-i,-i}$,

$$\begin{bmatrix} \mathcal{A}_S \\ \mathcal{A}_{\boldsymbol{\Xi}} \end{bmatrix}(\boldsymbol{\Lambda}_{-i,-i}) \;=\; c\,\begin{bmatrix} \mathrm{offdiag}(\mathbf{Q}_{-i,-i}^{(i)}) \\ (\mathbf{Q} - \mathbf{Q}^{(i)})_{-i,-i} \end{bmatrix} \;+\; \lambda_{ii}\,\begin{bmatrix} \mathrm{offdiag}\big(\boldsymbol{u}\,\mathbf{Z}_{i,-i}^{(i)} + \mathbf{Z}_{-i,i}^{(i)}\,\boldsymbol{u}^{\top}\big) \\ \boldsymbol{u}\,\boldsymbol{\Gamma}_{i,-i} + \boldsymbol{\Gamma}_{-i,i}\,\boldsymbol{u}^{\top} \end{bmatrix}.$$

**Assumption 3.** *The matrix $\boldsymbol{\Xi}^{-1}\mathbf{S}^{(i)}$ has a simple (pairwise-distinct) spectrum.*

Based on the above assumption and $\boldsymbol{\Xi}$ is generically invertible (see below), with the same argument used in the proof of single SCC, the stacked matrix uniquely identifies $\boldsymbol{\Lambda}_{-i,-i}$. Therefore, there exist data–dependent matrices $\mathbf{K}_0, \mathbf{K}_2$ (linear in the stacked right–hand sides) such that

$$\boldsymbol{\Lambda}_{-i,-i} = c\,\mathbf{K}_0 + \lambda_{ii}\,\mathbf{K}_2. \tag{39}$$

From equation 36,

$$\boldsymbol{\lambda}_{-i,i} = -\big(c\,\mathbf{K}_0 + \lambda_{ii}\mathbf{K}_2 + \lambda_{ii}\mathbf{I}\big)\,\boldsymbol{u}\,.$$

Using the *observational* $(i,-i)$ block of the Lyapunov equation,

$$\lambda_{ii}\mathbf{Z}_{-i,i} + \boldsymbol{\lambda}_{i,-i}\mathbf{Z}_{-i,-i} + \boldsymbol{\lambda}_{-i,i}Z_{ii} + \boldsymbol{\Lambda}_{-i,-i}\mathbf{Z}_{-i,i} = c\,\mathbf{Q}_{i,-i},$$

and $\mathbf{Z}_{-i,-i} \succ 0$, we get

$$\boldsymbol{\lambda}_{i,-i} = \big(c\,\mathbf{Q}_{i,-i} - \lambda_{ii}\mathbf{Z}_{-i,i} - \boldsymbol{\lambda}_{-i,i}Z_{ii} - \boldsymbol{\Lambda}_{-i,-i}\mathbf{Z}_{-i,i}\big)\,\mathbf{Z}_{-i,-i}^{-1}.$$

Substituting $\boldsymbol{\Lambda}_{-i,-i} = c\mathbf{K}_0 + \lambda_{ii}\mathbf{K}_2$ and $\boldsymbol{\lambda}_{-i,i} = -(c\mathbf{K}_0 + \lambda_{ii}\mathbf{K}_2 + \lambda_{ii}\mathbf{I})\boldsymbol{u}$ yields the affine form

$$\boldsymbol{\lambda}_{i,-i} = \mathbf{A}_0 + \lambda_{ii}\mathbf{A}_1\,,$$

where:

$$\mathbf{A}_0 = c\big(\mathbf{Q}_{i,-i}+Z_{ii}\,\boldsymbol{u}^\top\mathbf{K}_0^\top-\mathbf{Z}_{-i,i}\mathbf{K}_0^\top\big)\,\mathbf{Z}_{-i,-i}^{-1}\,, \quad \mathbf{A}_1 = \big(-\mathbf{Z}_{-i,i}+Z_{ii}\,\boldsymbol{u}^\top(\mathbf{I}+\mathbf{K}_2^\top)-\mathbf{Z}_{-i,i}\mathbf{K}_2^\top\big)\,\mathbf{Z}_{-i,-i}^{-1}.$$

Therefore, $\mathbf{A}_0$ scale linearly with the scale $c$; $\mathbf{K}_2$ and $\mathbf{A}_1$ do not involve parent terms and do not scale with $c$.

From the $i$-th component of the mean difference and the $(i,i)$ component of the Lyapunov difference, let $\Delta\mu_i := (\boldsymbol{\mu}_{\mathbf{T}|\mathbf{P}})_i - (\boldsymbol{\mu}_{\mathbf{T}|\mathbf{P}}^{(i)})_i$, $\Delta v_i := v_i^{(i)} - v_i$.Then

$$\lambda_{ii}\,\Delta\mu_i + \boldsymbol{\lambda}_{i,-i}(\boldsymbol{\mu}_{\mathbf{T}|\mathbf{P}})_{-i} = c\,\Delta v_i.$$

Insert $\boldsymbol{\lambda}_{i,-i} = \mathbf{A}_0 + \lambda_{ii}\mathbf{A}_1$ and group the terms to obtain

$$\lambda_{ii}\big(\Delta\mu_i + \mathbf{A}_1(\boldsymbol{\mu}_{\mathbf{T}|\mathbf{P}})_{-i}\big) = \big(c\Delta v_i - \mathbf{A}_0(\boldsymbol{\mu}_{\mathbf{T}|\mathbf{P}})_{-i}\big).$$

Hence the $\lambda_{ii}$ is identified as

$$\lambda_{ii} = \frac{c\Delta v_i - \mathbf{A}_0(\boldsymbol{\mu}_{\mathbf{T}|\mathbf{P}})_{-i}}{\Delta\mu_i + \mathbf{A}_1(\boldsymbol{\mu}_{\mathbf{T}|\mathbf{P}})_{-i}},$$

where it is determined up to the same scale $c$.

With $\lambda_{ii}$ identified, the previous equations give

$$\boldsymbol{\Lambda}_{-i,-i} = c\mathbf{K}_0 + \lambda_{ii}\,\mathbf{K}_2, \qquad \boldsymbol{\lambda}_{-i,i} = -\big(c\mathbf{K}_0 + \lambda_{ii}\,\mathbf{K}_2 + \lambda_{ii}\mathbf{I}\big)\boldsymbol{u}, \qquad \boldsymbol{\lambda}_{i,-i} = \mathbf{A}_0 + \lambda_{ii}\mathbf{A}_1,$$

so the entire $\boldsymbol{\Lambda}_{\mathbf{TT}}$ is determined with the same scaling $c$. Finally,

$$\mathbf{D}_{\mathbf{T}} = \mathrm{diag}\big(\boldsymbol{\Lambda}_{\mathbf{TT}}\mathbf{Z} + \mathbf{Z}\boldsymbol{\Lambda}_{\mathbf{TT}}^\top - \mathbf{Q}\big), \qquad \mathbf{b}_{\mathbf{T}} = \boldsymbol{\Lambda}_{\mathbf{TT}}\boldsymbol{\mu}_{\mathbf{T}|\mathbf{P}} + \mathbf{v}.$$

Thus $(\boldsymbol{\Lambda}_{\mathbf{TT}}, \mathbf{D}_{\mathbf{T}}, \mathbf{b}_{\mathbf{T}})$ are identified up to the same multiplicative scale $c$.

**Invertibility of $\boldsymbol{\Xi}$.** Let $\widetilde{\boldsymbol{\Lambda}}_{\mathbf{TT}}^{(i)} = \boldsymbol{\Lambda}_{\mathbf{TT}} - \mathbf{e}_i\mathbf{w}^\top, \mathbf{w} := \boldsymbol{\lambda}_{i,-i}$, where $\mathbf{w}$ collects the off–diagonal entries in row $i$. Then

$$\boldsymbol{\Lambda}_{\mathbf{TT}}\mathbf{Z} + \mathbf{Z}\,\boldsymbol{\Lambda}_{\mathbf{TT}}^\top = \mathbf{D}_{\mathbf{T}} + c\,\mathbf{Q}, \qquad \widetilde{\boldsymbol{\Lambda}}_{\mathbf{TT}}^{(i)}\mathbf{Z}^{(i)} + \mathbf{Z}^{(i)}\widetilde{\boldsymbol{\Lambda}}_{\mathbf{TT}}^{(i)\top} = \mathbf{D}_{\mathbf{T}} + c\,\mathbf{Q}^{(i)},$$

where $\mathbf{Q} = \mathbf{B}\bar{\mathbf{D}}_{\mathbf{P}}\mathbf{B}^\top$ and $\mathbf{Q}^{(i)} = \mathbf{B}^{(i)}\bar{\mathbf{D}}_{\mathbf{P}}\mathbf{B}^{(i)\top}$ are computable from data. Subtracting gives

$$\boldsymbol{\Lambda}_{\mathbf{TT}}\boldsymbol{\Gamma} + \boldsymbol{\Gamma}\,\boldsymbol{\Lambda}_{\mathbf{TT}}^\top = c\,(\mathbf{Q} - \mathbf{Q}^{(i)}) - \big(\mathbf{e}_i\mathbf{w}^\top\mathbf{Z}^{(i)} + \mathbf{Z}^{(i)}\mathbf{w}\mathbf{e}_i^\top\big), \qquad \boldsymbol{\Gamma} := \mathbf{Z} - \mathbf{Z}^{(i)}.$$

Consider the following representation for the Lyapunov equation,

$$\boldsymbol{\Gamma} = \int_0^\infty e^{-\boldsymbol{\Lambda}_{\mathbf{TT}}t}\,c(\mathbf{Q} - \mathbf{Q}^{(i)})\,e^{-\boldsymbol{\Lambda}_{\mathbf{TT}}^\top t}\,dt - \int_0^\infty e^{-\boldsymbol{\Lambda}_{\mathbf{TT}}t}\big(\mathbf{B}\mathbf{C} + \mathbf{C}^\top\mathbf{B}^\top\big)e^{-\boldsymbol{\Lambda}_{\mathbf{TT}}^\top t}\,dt,$$

with $\mathbf{B} := \mathbf{e}_i$ and $\mathbf{C} := \mathbf{w}^\top\mathbf{Z}^{(i)}$. Define the data–only quantities

$$\mathbf{u} := \frac{\mathbf{Z}_{-i,i}^{(i)}}{Z_{ii}^{(i)}}, \qquad \boldsymbol{\Xi} := \boldsymbol{\Gamma}_{-i,-i} - \mathbf{u}\,\boldsymbol{\Gamma}_{i,-i}.$$

Restrict to admissible parameters with $\boldsymbol{\Lambda}_{TP} = 0$. In this regime, the diffusion–difference term $c(\mathbf{Q} - \mathbf{Q}^{(i)})$ vanishes, so we need only consider the drift–difference term. The problem then reduces to a single SCC, with $\boldsymbol{\Xi}$ becoming exactly the one in 24. By Lemma 1, under Assumption 2, $\boldsymbol{\Xi}$ is generically invertible when $\boldsymbol{\Lambda}_{TP} = 0$. Moreover, $\det(\boldsymbol{\Xi})$ is an analytic function of the system parameters; since there exists a witness (at $\boldsymbol{\Lambda}_{TP} = 0$) for which $\det(\boldsymbol{\Xi}) \neq 0$, the determinant is not identically zero. Hence, for a generic choice of admissible parameters, $\boldsymbol{\Xi}$ is invertible.

## A.7 An Example with Multiple Root SCCs

Consider an OU process with state indices $1, 2, 3, 4$, where $1, 2, 3$ are root singletons and $4$ is a child singleton. Let

$$\mathbf{\Lambda} = \begin{bmatrix} \lambda_1 & 0 & 0 & 0 \\ 0 & \lambda_2 & 0 & 0 \\ 0 & 0 & \lambda_3 & 0 \\ a_1 & a_2 & a_3 & \lambda_4 \end{bmatrix}, \qquad \mathbf{D} = \mathrm{diag}(d_1, d_2, d_3, d_4),$$

and assume $\mathbf{D}$ is diagonal and each $\lambda_i > 0$. Suppose we observe the steady-state mean and covariance $(\boldsymbol{\mu}, \boldsymbol{\Sigma})$ and also the interventional moments $(\boldsymbol{\mu}^{(4)}, \boldsymbol{\Sigma}^{(4)})$ under an intervention on node $4$ that zeros the entries $a_i$s (please note that interventions on root give the same mean and covariance as the observational ones).

For each root $i \in \{1, 2, 3\}$, the scalar OU gives

$$\mu_i = \frac{b_i}{\lambda_i}, \qquad s_i := \Sigma_{ii} = \frac{d_i}{2\lambda_i}.$$

Let $t_i := \Sigma_{i4}$ denote the observed cross-covariances with the child. The off-diagonal Lyapunov equations yield

$$(\lambda_i + \lambda_4)t_i + s_i a_i = 0 \quad \Longleftrightarrow \quad a_i = -\frac{\lambda_i + \lambda_4}{s_i} t_i. \tag{40}$$

The child's observed mean and variance satisfy

$$\lambda_4 \mu_4 + \sum_{i=1}^{3} a_i \mu_i = b_4, \tag{41}$$

$$2\lambda_4 s_4 + 2 \sum_{i=1}^{3} a_i t_i = d_4. \tag{42}$$

Under the intervention on $4$ (which zeros the $a_i$s), we also observe

$$\mu_4^{(4)} = \frac{b_4}{\lambda_4}, \qquad 2\lambda_4 s_4^{(4)} = d_4, \qquad t_i^{(4)} = 0. \tag{43}$$

Now, fix any positive $(u_1, u_2, u_3)$ and a positive $v$ (to be chosen). Define

$$\lambda_i' = u_i \lambda_i, \quad b_i' = u_i b_i, \quad d_i' = u_i d_i \ (i = 1, 2, 3), \qquad \lambda_4' = v\lambda_4, \quad b_4' = vb_4, \quad d_4' = vd_4,$$

and choose $a_i'$ to preserve the observed $t_i$:

$$(\lambda_i' + \lambda_4')t_i + s_i a_i' = 0 \quad \Longleftrightarrow \quad a_i' = -\frac{u_i \lambda_i + v\lambda_4}{s_i} t_i. \tag{44}$$

Then for each root, $\mu_i' = \frac{b_i'}{\lambda_i'} = \mu_i$ and $s_i' = \frac{d_i'}{2\lambda_i'} = s_i$, while equation 44 ensures the same $t_i$.

For the interventional moments on $4$, scaling $(\lambda_4, b_4, d_4)$ by the common factor $v$ preserves $\mu_4^{(4)} = \frac{b_4}{\lambda_4}$ and $s_4^{(4)} = \frac{d_4}{2\lambda_4}$, and $t_i^{(4)} = 0$ still holds since the $a_i$ are zeroed under intervention. Thus all interventional moments match equation 43.

It remains to enforce that the primed parameters also satisfy the child's *observational* equations equation 41–equation 42. Subtracting $v$ times the original equation 42 from the primed version gives

$$2\lambda_4' s_4 - 2v\lambda_4 s_4 + 2\sum_{i=1}^{3}(a_i' - va_i)t_i = 0 \quad \Longleftrightarrow \quad \sum_{i=1}^{3}(a_i' - va_i)t_i = 0.$$

Using equation 40 and equation 44,

$$a_i' - va_i = -\frac{u_i \lambda_i + v\lambda_4}{s_i} t_i + v\frac{\lambda_i + \lambda_4}{s_i} t_i = \frac{(v - u_i)\lambda_i}{s_i} t_i.$$

Hence

$$\sum_{i=1}^{3}(v - u_i)\,\lambda_i\,\frac{t_i^2}{s_i} = 0 \iff v\sum_{i=1}^{3}\alpha_i = \sum_{i=1}^{3}u_i\alpha_i, \qquad \alpha_i := \lambda_i\frac{t_i^2}{s_i} > 0. \tag{45}$$

Thus

$$v = \frac{\sum_i u_i\alpha_i}{\sum_i \alpha_i}. \tag{46}$$

Similarly, subtracting $v$ times equation 41 from the primed version gives

$$\lambda_4'\mu_4 - v\lambda_4\mu_4 + \sum_{i=1}^{3}(a_i' - va_i)\mu_i = 0 \iff \sum_{i=1}^{3}(v - u_i)\,\lambda_i\frac{t_i}{s_i}\,\mu_i = 0,$$

i.e.,

$$v\sum_{i=1}^{3}\beta_i = \sum_{i=1}^{3}u_i\beta_i, \qquad \beta_i := \lambda_i\frac{t_i}{s_i}\mu_i = \frac{b_i t_i}{s_i}. \tag{47}$$

Thus also

$$v = \frac{\sum_i u_i\beta_i}{\sum_i \beta_i}. \tag{48}$$

Equating equation 46 and equation 48 yields a single linear constraint on $(u_1, u_2, u_3)$:

$$\frac{\sum_i u_i\alpha_i}{\sum_i \alpha_i} = \frac{\sum_i u_i\beta_i}{\sum_i \beta_i} \iff \sum_{i=1}^{3}u_i\Big(\alpha_i\sum_j\beta_j - \beta_i\sum_j\alpha_j\Big) = 0. \tag{49}$$

Let $\gamma_i := \alpha_i\sum_j\beta_j - \beta_i\sum_j\alpha_j$. Unless $(\gamma_1, \gamma_2, \gamma_3) = (0,0,0)$, equation 49 defines a nontrivial affine hyperplane in $\mathbb{R}^3$ (Note that the condition $\gamma_i = 0$ for all $i$ is nongeneric. Indeed, it requires $\frac{\beta_1}{\alpha_1} = \frac{\beta_2}{\alpha_2} = \frac{\beta_3}{\alpha_3}$, i.e., $\frac{b_i}{\lambda_i t_i}$ is constant across $i$. This imposes two independent algebraic constraints on the continuous parameters $(b_i, \lambda_i, t_i)$, hence holds only on a measure-zero subset of the parameter space). Therefore, the set of positive $(u_1, u_2, u_3)$ satisfying equation 49 has (at least) two degrees of freedom. For any such $(u_1, u_2, u_3)$, take $v$ from equation 46 (which equals equation 48) and define $a_i'$ by equation 44. Unless $u_1 = u_2 = u_3 = v$, the transformation is not a global rescaling of $(\mathbf{\Lambda}, \mathbf{b}, \mathbf{D})$, yet all moments (observational and interventional under node-4 intervention) coincide. Hence, identifiability up to a single global scale fails.

## B DETAILS OF EXPERIMENTS

### B.1 SYNTHETIC DATA

We generate synthetic datasets from a stable OU model. For each setup $|\mathcal{I}| \in \{0, 2, 4, 6, 8, 10\}$ (number of single-node interventions) and 40 instances, we fix $n = 10$ variables and sample:

- **Drift matrix $\mathbf{\Lambda}$**: start from zeros; for $i \neq j$, set $\Lambda_{ij} \sim \mathcal{N}(0.2, \sigma^2)$ with probability $\rho$ and 0 otherwise (default $\rho = 0.3$, $\sigma = 0.8$). Enforce positive stability by making each row strictly diagonally dominant:

$$\Lambda_{ii} \leftarrow \max\big(\texttt{DIAG\_MIN}, \sum_j |\Lambda_{ij}| + 0.2 + u_i\big), \quad u_i \sim \text{Unif}[0, 0.3], \ \texttt{DIAG\_MIN} = 0.8.$$

- **Bias vector and diffusion matrix**: $\mathbf{b} \sim \text{Unif}[0.2, 1.5]^n$; $\mathbf{D} = \text{diag}(\mathbf{d})$ with $d_k \sim \text{Unif}[0.2, 0.4]$.

Observational moments are computed via $\boldsymbol{\mu} = \mathbf{\Lambda}^{-1}\mathbf{b}$ and $\mathbf{\Lambda\Sigma} + \mathbf{\Sigma}\mathbf{\Lambda}^\top = \mathbf{D}$ (continuous-time Lyapunov). An intervention on node $k$ is implemented by zeroing row $k$ of $\mathbf{\Lambda}$ and restoring its diagonal entry (self-regulation preserved), leaving $\mathbf{b}$ and $\mathbf{D}$ unchanged; intervened moments $(\boldsymbol{\mu}^{(k)}, \mathbf{\Sigma}^{(k)})$ are computed analogously. We then draw snapshots from the corresponding Gaussians (by default 40,000 observational samples and 20,000 samples per intervention).

## B.2 Real Data

**Library-size normalization.** For each cell $k \in \{1, \dots, T_c\}$ in context/intervention $c$ with raw counts $\{x_k^i\}_{i=1}^N$ (genes $i = 1, \dots, N$), define the library size $s_k = \sum_{i=1}^N x_k^i$. We normalize by fractions $f_k^i = x_k^i / s_k$ and then scale to a common size

$$\tilde{x}_k^i \;=\; 10^4\, f_k^i \;=\; 10^4\, \frac{x_k^i}{s_k}.$$

All downstream statistics are computed on $\tilde{x}_k^i$. We denote the cell vector $\tilde{\mathbf{x}}_k = (\tilde{x}_k^1, \dots, \tilde{x}_k^N)^\top \in \mathbb{R}^N$.

**Empirical moments with shrinkage.** Given a context $c$ with $T_c$ cells $\{\tilde{\mathbf{x}}_k\}_{k=1}^{T_c}$, the empirical mean and (shrunk) covariance are

$$\hat{\boldsymbol{\mu}}_c = \frac{1}{T_c} \sum_{k=1}^{T_c} \tilde{\mathbf{x}}_k, \qquad \hat{\boldsymbol{\Sigma}}_c = (1 - \eta_c)\, \hat{\boldsymbol{\Sigma}}_{c,\text{sample}} + \eta_c\, \text{diag}(\hat{\boldsymbol{\Sigma}}_{c,\text{sample}}),$$

where $\hat{\boldsymbol{\Sigma}}_{c,\text{sample}} = \frac{1}{T_c - 1} \sum_{k=1}^{T_c} (\tilde{\mathbf{x}}_k - \hat{\boldsymbol{\mu}}_c)(\tilde{\mathbf{x}}_k - \hat{\boldsymbol{\mu}}_c)^\top$ and $\eta_c = \min\big(0.3,\ 10/(T_c - 1)\big)$.

**Model.** We fit an OU model with parameters $(\boldsymbol{\Lambda}, \mathbf{b}, \boldsymbol{D})$ on wild-type and interventional data. For an intervention on gene $k$, the context-specific drift is obtained from $\boldsymbol{\Lambda}$ by zeroing all off-diagonal entries in row $k$ and keeping $\Lambda_{kk}$; $\mathbf{b}$ and $\boldsymbol{D}$ are shared. We enforce $\text{diag}(\boldsymbol{\Lambda}) > 0$ (softplus) and rescale to $\text{tr}(\boldsymbol{\Lambda}) = n$.

**Train/test split and metrics.** Similar to previous work Rohbeck et al. (2024), interventions are split into train context IDs $\{0, \dots, 54\}$ and held-out test context IDs $\{55, \dots, 60\}$. Evaluation uses the same per-cell normalization on two metric DES and PDS.

## B.3 Effect of Sample Size on Relative Error

To assess the sensitivity of our estimator to the number of steady-state samples, we conducted an experiment in which we varied the number of observational samples used to estimate $(\hat{\boldsymbol{\mu}}, \hat{\boldsymbol{\Sigma}})$. For each trial, the number of samples per intervention was set to half of the observational sample size. We evaluated the relative error of recovering $\boldsymbol{\Lambda}$ for observational sample sizes $\{5k, 10k, 20k, 40k\}$ over 30 random instances with $n = 10$. As shown in Fig. 2, the estimation error decreases markedly when increasing the number of samples from 5k to 20k and then stabilizes, indicating that the estimator achieves accurate recovery with a moderate number of steady-state samples. This behavior is expected, since both empirical means and covariances become sufficiently accurate in this regime, after which additional samples yield diminishing returns.

# C About the Spectral Assumptions

## C.1 Numerical Analysis of Spectral Assumptions

We assessed the spectral nondegeneracy assumptions by Monte Carlo analysis on random, strongly connected OU models. In each of 250 trials, we sampled a drift matrix $\boldsymbol{\Lambda} \in \mathbb{R}^{10 \times 10}$ with off–diagonal pattern drawn i.i.d. at density $0.3$ (Gaussian weights, standard deviation $0.8$), and set each diagonal entry strictly larger than the $\ell_1$ row sum (ensuring Hurwitz stability). Strong connectivity was checked on the directed graph with edge $j \to i$ iff $\Lambda_{ij} \neq 0$. We drew a diagonal diffusion $\mathbf{D}$ with $d_k \sim \text{Unif}[0.2, 0.4]$, selected an intervention coordinate $i$ uniformly at random, and formed the intervened drift by zeroing row $i$'s off–diagonals (keeping $\Lambda_{ii}$). We then solved the continuous–time Lyapunov equations to obtain the observational and intervened covariances $\boldsymbol{\Sigma}$ and $\boldsymbol{\Sigma}^{(i)}$, formed $\boldsymbol{\Gamma} := \boldsymbol{\Sigma} - \boldsymbol{\Sigma}^{(i)}$, and built the $(-i, -i)$ Schur blocks $\boldsymbol{\Xi}$ and $\mathbf{Z}$ used in the proofs, with $\mathbf{A} := \boldsymbol{\Xi}^{-1} \mathbf{Z}$. We evaluated: (i) *simple spectrum* for $\boldsymbol{\Lambda}$ and $\mathbf{A}$ via the minimum pairwise eigenvalue spacing; (ii) condition on $\mathbf{W}_\Lambda$ from Assumption 2 via the minimum distance of $\sigma(\mathbf{W}_\Lambda^{-T} \mathbf{W}_\Lambda)$ to

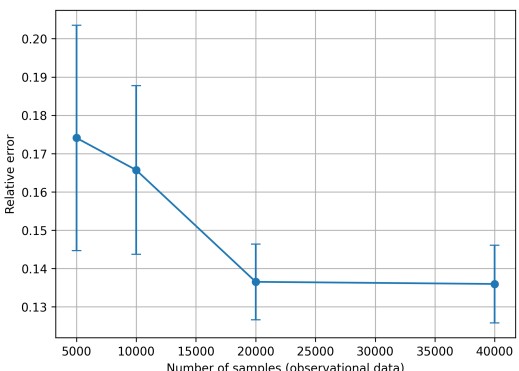

Figure 2: The average relative error for recovering $\Lambda$ (up to some global scaling) versus the sample size (observational data).

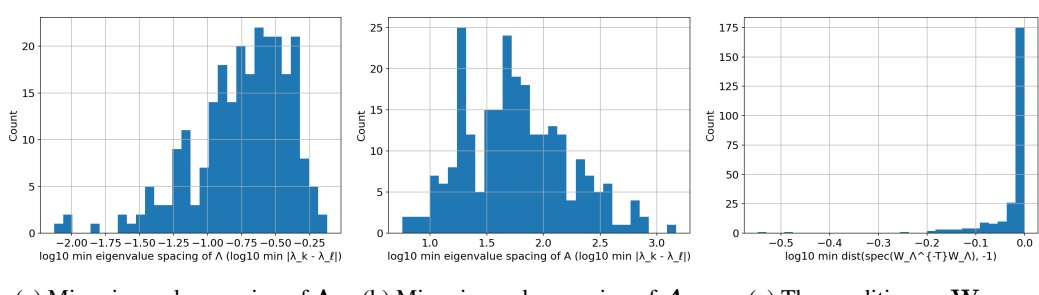

(a) Min. eigenvalue spacing of $\Lambda$  (b) Min. eigenvalue spacing of $A$  (c) The condition on $\mathbf{W}_\Lambda$

Figure 3: Numerical probe of spectral assumptions: a) The histogram of minimum pairwise eigenvalue spacing of $\Lambda$. b) The histogram of minimum pairwise eigenvalue spacing of $A$. c) The condition on $\mathbf{W}_\Lambda$ from Assumption 2 (The values on the x-axis are in $\log_1 0$ scale)

$-1$. We report the results in Figure 3. As can be seen, in all the trials, the three spectral assumptions hold.

### C.2 ABOUT THE ROLE OF SPECTRAL ASSUMPTIONS IN THE PROOFS

- Assumption 1 (simple spectrum of $A := \Xi^{-1}Z$) is used in the proof of Theorem 1 (single SCC). In the single-SCC case, we obtain two Lyapunov-type equations for the unknown block $(\Lambda_{-i,-i}, D_{-i,-i})$ and reduce them to a constraint of the form $SA - A^\top S$ being diagonal, where $S := \Lambda_{-i,-i}\Xi$ and $A := \Xi^{-1}Z$. Assumption 1 rules out the degenerate case where $A$ has repeated eigenvalues; under simple spectrum, the only skew-symmetric $S$ compatible with this constraint is $S = 0$, which forces $D_{-i,-i}$ and the off-diagonal part of $\Lambda_{-i,-i}$ to be uniquely determined by the moments.

- Assumption 2 is also used in the proof of Theorem 1. It has two parts: (i) $\Lambda$ has simple spectrum, and (ii) a nondegeneracy condition on the matrix $W_\Lambda$ built from the eigenvectors of $\Lambda$. This assumption ensures that the difference covariance $\Gamma := \Sigma - \Sigma^{(i)}$ leads to an invertible Schur block $\Xi$ (Lemma 1).

- Assumption 3 (simple spectrum of $\Xi^{-1}S^{(i)}$) appears in the proof of Theorem 3 (multi-SCC case). After conditioning on previously resolved components $P$, the remaining block $T$ leads to a stacked linear system for $\Lambda_{TT}$ built from two Lyapunov-type equations. Assumption 3 is the multi-SCC analogue of Assumption 1. It rules out spectral degeneracies of $\Xi^{-1}S^{(i)}$ so that this stacked system is injective and determines the drift within the target block $T$ (up to the global scaling).

# D    USE OF LARGE LANGUAGE MODELS

We used a large language model (LLM) primarily to polish the writing in the main text and to improve the presentation of equations in the appendix. In addition, the LLM was used to verify the correctness of proofs and to assist with coding and debugging in the experiments.

