# OpenReview forum: "One Intervention per Component is Enough: Towards Identifiability in Linear Stochastic Dynamics from Steady State"
_ICLR.cc/2026/Conference — Submitted to ICLR 2026_

### Official Review · Reviewer_MWjF · 2025-10-17

**Soundness:** 3
**Presentation:** 3
**Contribution:** 2
**Rating:** 4
**Confidence:** 2

**Summary:**

The paper analyzes identifiability of linear stochastic dynamics (Ornstein–Uhlenbeck processes) from steady-state observational and interventional data.
It proves that one intervention per strongly connected component (SCC) suffices to identify the system up to a global scale under a “row-zeroing” intervention model and diagonal diffusion.
A regularized least-squares estimator is proposed, with synthetic and Perturb-seq experiments illustrating the approach.

**Strengths:**

•	Clear theoretical framing and well-presented identifiability proofs.
	•	The “one intervention per SCC” condition is simple and interpretable.
	•	Empirical evaluation includes both synthetic and real (Perturb-seq) datasets.

**Weaknesses:**

•	The intervention model (zeroing off-diagonals of one row while keeping the diagonal) is restrictive (independent dynamics) and not biologically realistic.
	•	The estimator is a straightforward least-squares fit; no methodological innovation beyond the theoretical analysis.
	•	No comparison to existing system-identification or Optimal Experimental Design (OED) approaches
	•	Focused entirely on linear OU dynamics, limiting relevance to broader ML audiences.
	•	Overall contribution is more aligned with causal inference and system identification than with core machine-learning or representation-learning research, not as relevant to the ICLR community.

**Questions:**

1.	How sensitive is the identifiability result to imperfect or partial interventions?
	2.	How can extensions be handled for non-diagonal diffusion or nonlinear dynamics?
	3.	Have you compared against classical gradient-based or Bayesian OED estimators using the same steady-state moments?

---

> ### Author Response · Authors · 2025-11-20
> **Response**
>
> > The intervention model (zeroing off-diagonals of one row while keeping the diagonal) is restrictive (independent dynamics) and not biologically realistic.
>
> We agree that perfect interventions are an idealization, especially in biological systems where perturbations can have off-target effects. However, the notion of hard interventions is standard in the causal SDE literature, see, e.g., Hansen and Sokol (2014) and Boeken and Mooij (2024), where some post-intervention SDEs are formulated in this idealized setting. In the context of gene regulatory networks, semi-synthetic datasets often model gene knockouts as such hard interventions: for example, recent work based on SERGIO simulator implements knockouts by forcing the production rate of the intervened gene to zero (Hägele et al., 2023), and gene knockouts are typically treated as targeted interventions in causal-discovery surveys (e.g., Brouillard et al., 2024).
>
> > The estimator is a straightforward least-squares fit; no methodological innovation beyond the theoretical analysis.
>
> The main contribution of the paper is the identifiability result, and the estimator is designed to mirror the moment equations used in the theory. In particular, the Lyapunov-residual part of the loss has appeared in prior work, but our analysis shows that incorporating the steady-state mean equations can be used in identifying the SCCs. Empirically, we find that adding these mean terms substantially improves parameter recovery in synthetic and real data. Thus, the empirical results show the practical value of including the mean constraints.
>
> > No comparison to existing system-identification or Optimal Experimental Design (OED) approaches
>
> To the best of our knowledge, most existing system-identification methods for dynamical systems assume access to full trajectories, rather than only snapshot samples from the stationary distribution, which is the setting we focus on here. Moreover, optimal
> experimental design has been developed primarily in the context of structural causal models (most often, having an acyclicity assumption) rather than for SDEs identified from steady-state data.
>
> > Focused entirely on linear OU dynamics, limiting relevance to broader ML audiences.
>
> While our theoretical results are developed for linear OU processes, we note that this setting is a
> canonical model class for studying causal inference and perturbation prediction in continuous-time
> systems. The identifiability results we obtain lead to some practical insights: (i) they characterize when steady-state interventions are sufficient
> to recover causal structure, and (ii) they enable prediction of unseen
> perturbations. More broadly, the OU setting offers a tractable
> framework in which causal effects, intervention semantics, and parameter recovery can be analyzed
> rigorously. These insights are valuable for the wider ML community working on causal inference and perturbation
> prediction.
>
> > Overall contribution is more aligned with causal inference and system identification than with core machine-learning or representation-learning research, not as relevant to the ICLR community.
>
> We respectfully disagree that the contribution falls outside the scope of ICLR. Causal inference and causal discovery have become central themes at ICLR in recent years. For instance, please check the 2024 oral paper “Gene Regulatory Network Inference in the Presence of Dropouts: a Causal View” (openreview.net/forum?id=gFR4QwK53h) which shows the community’s strong interest in causal structure learning.
>
> > Regarding questions
>
> Our current identifiability analysis relies on perfect (hard) interventions and diagonal diffusion; extending the theory to imperfect or partial interventions, non-diagonal diffusion, or nonlinear dynamics would require substantially new techniques. We view all of these
> directions as important avenues for future work.
>
> Boeken, Philip, and Joris M. Mooij. "Dynamic structural causal models." arXiv preprint arXiv:2406.01161 (2024).
>
> Hansen, Niels, and Alexander Sokol. "Causal interpretation of stochastic differential equations." (2014): 1-24.
>
> Hägele et al. "Bacadi: Bayesian causal discovery with unknown interventions." International Conference on Artificial Intelligence and Statistics. PMLR, 2023.
>
> Brouillard et al. "The landscape of causal discovery data: Grounding causal discovery in real-world applications." arXiv preprint arXiv:2412.01953 (2024).

---

### Official Review · Reviewer_iMRJ · 2025-10-27

**Soundness:** 3
**Presentation:** 2
**Contribution:** 2
**Rating:** 4
**Confidence:** 3

**Summary:**

The paper studies identifiability of linear multivariate OU dynamics from steady-state observational + interventional snapshots. Its core result: one intervention per strongly connected component (SCC) identifies parameters up to a global scale when the SCC condensation graph is connected with a single root and mild spectral non-degeneracy holds. The proofs tie directly to a practical estimator that jointly fits means and covariances. Experiments on synthetic and single-cell data show sensible recovery and prediction for unseen interventions.

**Strengths:**

1. **Well-motivated problem.**
The paper targets the steady-state snapshot regime, where trajectories are unavailable, which is common and practical in applications such as single-cell perturbation screens.
2. **Concise identifiability guarantee.**
It shows that a single intervention per SCC (under a single-root condensation DAG) is sufficient to identify parameters up to a global scale, relaxing prior requirements that often needed interventions on many or most nodes.

**Weaknesses:**

1. **Missing basic definitions in the main text.**
Key notions: e.g., strongly connected component (SCC) and steady-state mean/covariance, are not clearly defined where they first appear. A brief one-sentence explanation for each would improve accessibility for readers who are not specialists in stochastic dynamics.

2. **Assumptions are hard to map to results.**
Assumptions 1–3 are introduced in the appendix, making it difficult to see how each supports a specific theorem. Providing a short, intuitive explanation next to each theorem (what the assumption rules out and why it is needed) would significantly aid comprehension.

3. **Main-text proof density.**
The detailed proof of Theorem 3 occupies a full page and renders the exposition equation-heavy. Consider moving the derivation to the appendix and keeping a high-level proof sketch in the main text.

4. **Restrictive diffusion model.**
The assumption of diagonal diffusion is strong. Many real systems exhibit correlated noise. Some discussion of robustness or extensions would be helpful.

**Questions:**

1. **Generic vs. strong assumptions.**
Which assumptions are intended to be generic (holding except on a measure-zero set) and which are strong modeling or structural? Please add brief intuitive explanations for each assumption and, if feasible, a theoretical discussion of genericity (beyond the numerical evidence in Appendix C).

2. **Resolving the global scale.**
The identifiability result is up to a global scaling factor. For applications requiring absolute parameters, what concrete strategies do you recommend to fix the scale (e.g., anchoring a known rate/variance or using an external calibration)?

---

> ### Author Response · Authors · 2025-11-20
> **Response - Part 1**
>
> > Missing basic definitions in the main text...
>
> We added the definition of SCC in Appendix A.1\. We also explicitly defined $\mathbf{x}_{\infty},$ which denotes a random vector distributed according to the stationary distribution of the OU process. With that, the definitions of mean and covariance are more clear now (which were given in line 91 and line 97 in the original submission).
>
> > Assumptions are hard to map to results. Assumptions 1–3 are introduced in the appendix, making it difficult to see how each supports a specific theorem.
>
> All three assumptions are technical spectral nondegeneracy conditions that are used to guarantee the uniqueness of solutions to certain linear systems that arise in the proofs; they are not structural conditions on the graph itself. Concretely:
>
> - Assumption 1 (simple spectrum of $A := \Xi^{-1}Z$) is used in the proof of Theorem 1 (single SCC). In the single-SCC case, we obtain two Lyapunov-type equations for the unknown block $(\Lambda_{-i,-i}, D_{-i,-i})$ and reduce them to a constraint of the form $SA - A^\top S$ is diagonal, where $S := \Lambda_{-i,-i}\Xi$ and $A := \Xi^{-1}Z$. Assumption 1 rules out the degenerate case where $A$ has repeated eigenvalues; under simple spectrum, the only skew-symmetric $S$ compatible with this constraint is $S=0$, which forces $D_{-i,-i}$ and the off-diagonal part of $\Lambda_{-i,-i}$ to be zero.
>
> - Assumption 2 is also used in the proof of Theorem 1. It has two parts: (i) $\Lambda$ has simple spectrum, and (ii) a nondegeneracy condition on the matrix $W_\Lambda$ built from the eigenvectors of $\Lambda$. This assumption ensures that the difference covariance $\Gamma := \Sigma - \Sigma^{(i)}$ leads to an invertible Schur block $\Xi$ (Lemma 1).
>
> - Assumption 3 (simple spectrum of $\Xi^{-1}S^{(i)}$) appears in the proof of Theorem 3 (multi-SCC case). After conditioning on previously resolved components $P$, the remaining block $T$ leads to a stacked linear system for $\Lambda_{TT}$ built from two Lyapunov-type equations. Assumption 3 is the multi-SCC analogue of Assumption 1. It rules out spectral degeneracies of $\Xi^{-1}S^{(i)}$.
>
> We emphasize that we do not currently provide formal genericity proofs for these spectral assumptions; instead, we assessed them numerically in Appendix C and observed that they hold with measure one for random strongly connected OU models. In the revision, we will keep the formal statements of Assumptions 1–3 in the appendix. Remark 1 already briefly discusses their role. In Appendix C.2, we added the more detailed explanations (as outlined above) to further clarify these assumptions.
>
> >  Main-text proof density. The detailed proof of Theorem 3 occupies a full page and renders the exposition equation-heavy...
>
> We appreciate the reviewer’s suggestion. We agree that the full proof of Theorem 3 is technical, and for this reason, we have already moved the more technical parts to Appendix A.6. The part in the main text explains how conditioning on previously resolved components, reduces the multi-SCC case to a sequence of single-SCC problems. This outline is essential for understanding the design and rationale of Algorithm 1, which mirrors the structure of the proof. For this reason, we prefer to keep the
> current level of detail in the main text while maintaining the technical derivations in the appendix.
>
> > Restrictive diffusion model. The assumption of diagonal diffusion is strong...
>
> We agree that assuming a diagonal diffusion matrix is a modeling restriction and that real systems could exhibit correlated noise. We followed much of the existing literature on learning linear SDEs from stationary data, where diagonal diffusion is standard, e.g.,
> Varando and Hansen (2020), Dettling et al. (2023), and Rohbeck et al. (2024). Conceptually, this assumption is analogous to the causal sufficiency assumption in the SCM framework where non-zero off-diagonal entries in the diffusion would correspond to shared noise sources and hence to latent confounding between variables. Extending our identifiability results and learning procedures to settings with such latent confounding is an important direction for future work.
>
> > Generic vs. strong assumptions. Which assumptions are intended to be generic (holding except on a measure-zero set) and which are strong modeling or structural?...
>
> The structural assumptions in Theorem 3 (namely that the condensation graph over SCCs is connected and has a single root
> SCC) are in fact necessary conditions for identifiability (see Remark 3). If these conditions fail, no method using only stationary observational/interventional distributions can recover the parameters of linear SDE up to a single global scaling. The spectral assumptions are generic. These assumptions rule out algebraic degeneracies (e.g., repeated eigenvalues).

---

> ### Author Response · Authors · 2025-11-20
> **Response - Part 2**
>
> > Resolving the global scale. The identifiability result is up to a global scaling factor...
>
> Our identifiability result is indeed up to a single global scaling factor, which reflects the fact that steady-state moments alone cannot fix the scaling of the OU parameters. In many applications, this is sufficient (e.g., for predicting intervention effects on steady-state means and covariances), but when absolute parameters are required, one can fix the scale by anchoring known rates. For example, if prior knowledge is available about the typical range of self-regulatory rates (diagonal entries of $\Lambda$) for a subset of nodes, one can constrain $\Lambda_{ii}$ to a known value or interval and rescale all parameters accordingly.

---

### Official Review · Reviewer_zTTa · 2025-10-28

**Soundness:** 3
**Presentation:** 2
**Contribution:** 2
**Rating:** 4
**Confidence:** 3

**Summary:**

This paper studies the identifiability of linear stochastic dynamical systems (multivariate OU processes) when only snapshot data is available under both observational and interventional conditions. The paper proves that one intervention per strongly connected component (SCC) of the underlying graph is sufficient to identify the entire system up to a single global scaling factor. They also provide a recursive algorithm that learns the SCC structure from interventional mean shifts and recovers parameters within each SCC using steady-state mean and covariance equations jointly. A practical regularized least-squares objective is proposed to fit all observed (interventional and observational) moments and tested in synthetic and real-world experiments.

**Strengths:**

- the task of recovering causal dynamics from steady-state data is very relevant in scientific and machine learning settings
- the rule behind the one intervention per SCC is clean and simple, offering practical guidance for experimental design
- the theoretical results are rigorous and supported by synthetic and real-data experiments
- the focus on "snapshot" data fits many real-world applications, such as biology, where temporal measurements are often available

**Weaknesses:**

- since in this linear setting identifiability directly corresponds to causal discovery, the related work section could better emphasize this by referring to similar works in the field. See questions below.
- the paper refers to strongly connected components (SCCs) but it seems that it does not provide a definition. I would recommend adding it
- perfect interventions is not really realistic in a real-world setting, especially in biological perturbations. I think that the theoretical results are interesting, however I am not sure how insightful the real-world experiments are. This learning framework is based on strong modeling assumptions (linear SDE, diagonal diffusion, perfect interventions) which are likely not met in these applications. Perhaps this could be mentioned by the authors.

**Questions:**

- I am missing the link to these references on causal discovery within dynamical systems. Can the authors elaborate on these? For causal discovery of SDEs [1,2] and with interventions [3,4].
- how can these results be used for practical guidance in experimental design if one does not know a priori which ones are the SCC?
- Remark 1 states that spectral nondegeneracy assumptions hold generically within SCCs, however how does this extend to sparse settings?

[1] Boeken, Philip, and Joris M. Mooij. "Dynamic structural causal models."

[2] Manten, Georg, et al. "Signature kernel conditional independence tests in causal discovery for stochastic processes." (2024).

[3] Hansen, Niels, and Alexander Sokol. "Causal interpretation of stochastic differential equations." (2014).

[4] Zweig, Aaron, et al. "Towards Identifiability of Interventional Stochastic Differential Equations." (2025).

---

> ### Author Response · Authors · 2025-11-20
> **Response - Part 1**
>
> > Since in this linear setting identifiability directly corresponds to causal discovery, the related work section could better emphasize this by referring to similar works in the field...
>
> We thank the reviewer for pointing out these works. We focused the related work section on methods that, mainly operate based on stationary data (not observing the whole trajectory).
>
> Boeken and Mooij and Manten et al. studied causal discovery for stochastic processes based on full or partial trajectories and conditional independence constraints on path space. Their goal is to recover causal graphs from time-series data, whereas we assume only observing steady-state samples under a limited number of interventions. Hansen and Sokol provided a causal semantics for SDEs and show that post-intervention distributions are identifiable from the generator under Levy noise.
>
> Zweig et al. is the closest work to our setting, as they also study identifiability of SDEs from stationary distributions where interventions are modeled by adding a vector to the drift (i.e., a mean intervention, rather than the hard
> interventions we considered here). Their analysis assumes a low-rank parameterization of the drift, $v(x) = (AB - D)x$ with $A \in \mathbb{R}^{n \times r}$, $B \in \mathbb{R}^{r \times n}$, so that
> the effective drift $AB$ has rank $r \ll n$, and their bounds on the number of required interventions scale with this latent dimension $r$. In contrast, we study general drift matrices, and our identifiability condition is
> expressed in terms of the SCC structure of the drift graph, and we provide a SCC-wise recovery algorithm based on steady-state means and covariances. We added a summary of this explanation to the related work.
>
> > The paper refers to strongly connected components (SCCs) but it seems that it does not provide a definition.
>
> A strongly connected component (SCC) of a directed graph $G$ is a maximal subset of vertices $C \subseteq V(G)$ such that for any $u, v \in C$ there exists a directed path from $u$ to $v$. The SCCs of $G$ form a partition of its nodes. We added a graph definition section in Appendix A.1.
>
> > Perfect interventions is not really realistic in a real-world setting, especially in biological perturbations.
>
> We agree that perfect interventions are an idealization, especially in biological systems where perturbations can have off-target effects. However, the notion of hard interventions is standard in the causal SDE literature, see, e.g., Hansen and Sokol (2014) and Boeken and Mooij (2024), where some post-intervention SDEs are formulated in this idealized setting. In the context of gene regulatory networks, semi-synthetic datasets often model gene knockouts as such hard interventions. For example, recent work based on SERGIO simulator implements knockouts by forcing the production rate of the intervened gene to zero (Hägele et al., 2023), and gene knockouts are typically treated as targeted interventions in causal discovery literature (e.g., Brouillard et al., 2024).
>
> Regarding the linear SDE with diagonal diffusion, we also acknowledge that this is a simplifying assumption for real-world biology applications. Our motivation is twofold: (i) from a theoretical perspective, identifiability from a limited number of interventions is already non-trivial in this linear setting, and the graphical conditions on the drift matrix can be analyzed explicitly; (ii) from a practical perspective, when the number of samples per intervention is limited, linear models are typically much more sample-efficient. Moreover, they are also more interpretable than deep models, since the drift matrix directly encodes causal relationships, whereas deep or nonlinear models tend to be more black-box. We will make these modeling choices and their limitations explicit in the revised version.
>
> > How can these results be used for practical guidance in experimental design if one does not know a priori which ones are the SCC?
>
> Theorem 2 shows how, from a given intervention, we can learn some ancestral relationships among the variables in the system. Concretely, for an intervention on node $i$, $R_i$ denotes the set of variables whose steady-state mean changes. Theorem 2 implies that, generically, $R_i$ shows the set of variables that variable $i$ has a direct path to.
>
> From an experimental-design perspective, this suggests a simple adaptive strategy when the SCCs are unknown. Starting from a given initial interventions, one can (i) compute the sets $R_i$, (ii) group nodes with identical $R_i$ to obtain candidate SCCs, and (iii) identify unexplored regions of the graph as those nodes whose means never change. New interventions can then be targeted to such nodes in order to recover additional candidate SCCs.

---

> ### Author Response · Authors · 2025-11-20
> **Response - Part 2**
>
> > Remark 1 states that spectral nondegeneracy assumptions hold generically within SCCs, however how does this extend to sparse settings?
>
> Within an SCC, the sparsity pattern of the dirft matrix is not arbitrary. By definition, every node has a directed path to every other node. While we do not provide a formal genericity proof, our numerical evidence in Appendix C shows that the spectral conditions are generically satisfied for strongly connected components.
>
> Hägele et al. "Bacadi: Bayesian causal discovery with unknown interventions." International Conference on Artificial Intelligence and Statistics. PMLR, 2023.
>
> Brouillard et al. "The landscape of causal discovery data: Grounding causal discovery in real-world applications." arXiv preprint arXiv:2412.01953 (2024).

---

> ### Comment · Reviewer_zTTa · 2025-11-27
> **Thank you**
>
> I thank the authors for the reply, they have addressed most of my questions. I will increase the score accordingly.

---

### Official Review · Reviewer_WHmi · 2025-10-30

**Soundness:** 4
**Presentation:** 3
**Contribution:** 3
**Rating:** 6
**Confidence:** 3

**Summary:**

The paper studies the problem of recovering the parameters of a causal linear system given by a multivariate Ornstein-Uhlenbeck (OU) process from steady-state observational and interventional data. The drift matrix \Lambda is the key parameter of the OU process that differentiates observational and interventional data. Observational data for the $i$th random variable (RV) is sampled from an underlying OU process. Interventional data of $i$th RV is obtained from a modified OU process driven by $\tilde \Lambda^{(i)}$, which isolates $i$ from its regulators ($j\neq i$). The OU parameters are recovered from observational and interventional first- and second-order moments $\mu$ and $\Sigma$, and $\tilde \mu$ and $\tilde \Sigma$. The paper is theory-focused. A (drift) graph $\mathcal{G}(\Lambda)$ is formed by non-zero entries of the drift matrix $\Lambda$. The authors prove that under some conditions, if $\mathcal{G}$ has $N$ strongly connected components (SCCs), then the number of interventions necessary to estimate the parameters of OU up to a scaling factor equals $N$ (one intervention per SCC). Based on the analysis, the authors propose a least-squares estimator of OU parameters. Empirical simulations on real and synthetic datasets demonstrate good performance and corroborate theoretical results.

**Strengths:**

- The paper conducts a thorough theoretical analysis of how interventions aid the estimation of OU parameters
- The estimator is developed from the theoretical analysis, quantifying the information needed to estimate the parameters
- It is well-written, supported by a number of recent publications
- Much of prior work focused on either identifiability of OU parameters or interventions to learn stationary stochastic differential equations (SDE) models. Only very recent work (Rohbeck’24) combined both. The authors build on that work and provide identifiability results under weaker assumptions.
-  Beyond the theoretical identifiability results, the paper contributes a concrete estimation algorithm that operationalizes the theory. The proposed regularized least-squares estimator jointly fits the steady-state mean and covariance equations under observational and interventional data, incorporating sparsity priors on the drift matrix. This bridges the gap between abstract identifiability proofs and practical parameter recovery. The algorithm is also evaluated on both synthetic and real biological datasets (Perturb-seq), demonstrating that the theoretical insights translate into tangible performance gains and confirming the model’s robustness in realistic, noisy settings.

**Weaknesses:**

- Insufficient number of baselines. The authors reference a number of relevant works. However, both experiments compare only against a single baseline, which is “similar” to prior work. Hence, it is difficult to evaluate the relative performance. In my opinion, it would be helpful to compare against traditional and non-interventional estimators.
- Lack of empirical support. The main claim is that we only need a single intervention per SCC. However, in the experiment on the synthetic dataset, the authors only analyze whether the drift graph meets conditions from Thm. 3. What about generating a graph with $N$ SCCs and showing that, on average, $N$ interventions (one per SCC) are needed for a good estimate? For the number of SCCs smaller than $N$, the error is expected to increase, while for larger numbers, it would marginally decrease. One could also add directed two-cycles to confirm that the gain w.r.t. the work of Dettling’23 is even more significant in this case.
- Minor clarity comments. One or two sentences in the main text describing what the topological order of SCCs intuitively means (Remark 2) would help. Also Intervention definition (l. 107) should be $\Lambda_{kj}$ in the first row. Lastly, $x_\inf$ (l. 092) and $\oplus$ (l. 215) are undefined.

**Questions:**

- (10) assumes that we have access to observational and interventional moments. How sensitive is this model to the measurement errors? What if some of the parameters (e.g., diffusion matrix) are misspecified?
- An experiment on a real dataset states that the scaling factor cancels out when estimating the first-order moment $\mu$. But would it also cancel when estimating the covariance $\Sigma$? If it does not cancel out, what are the practical implications of this scaling ambiguity?
- Which assumptions from Thm. 3 can be relaxed without losing identifiability?

---

> ### Author Response · Authors · 2025-11-20
> **Response - Part 1**
>
> > Insufficient number of baselines. The authors reference a number...
>
> We appreciate the reviewer’s suggestion to include more baselines. Our main contribution is theoretical, which is providing identifiability results for linear stochastic differential equations from steady-state data (Theorems 1-3). The empirical section is meant to illustrate the practical implications of this identifiability result rather than to benchmark a new learning algorithm.
>
> Regarding comparing with deep models (e.g., transformer-based perturbation predictors), while they might achieve higher predictive accuracy, herein, we mainly focus on comparing with the work considering linear SDEs using steady-state data. In particular, Rohbeck et al. (2024) used a part of our loss without the mean terms in their loss. We already include this as an ablation (“Covariance only”). Varando & Hansen (2020) proposed an $l_1$-regularized estimator from observational covariances only (no interventions) and therefore it is not designed for our interventional setting. Lorch et al. (2024) also proposed a version of their loss for linear SDEs, but their interventions are implemented as mean shifts rather than the hard interventions considered in our work. We adapted their loss to our hard-intervention setting and evaluated it on our synthetic data. So far, we have not observed a decrease in relative error as the number of interventions increases. We will perform additional fine-tuning during the discussion phase to see whether their performance can be improved.
>
> > Lack of empirical support. The main claim is that...
>
> We emphasize that our identifiability result provides sufficient (not necessary) conditions. Thus, there may exist linear SDEs whose parameters are identifiable up to a global scaling even if they do not satisfy the graphical conditions in Theorem 3. Nevertheless, we constructed a structure of five SCCs, each consisting of a two-node cycle, and with a linear condensation graph obtained by connecting each component to the next through a randomly selected node. Over 30 such instances and for $\lvert \mathcal{I} \rvert = k$ interventions, we checked in how many cases the OU parameters were identifiable up to a global scaling. We observed the following: for $k=0$, identifiability fails in 30/30 cases; for $k=1$, it fails in 11/30 cases; for $k=2$, it fails in 4/30 cases; and for $k=3$, it fails in 0/30 cases. While one might expect identifiability in all instances to occur later (e.g., around $k=5$-$6$), this experiment illustrates that the conditions are sufficient but not necessary.
>
> > Minor clarity comments.
>
> Because the condensation graph (with nodes corresponding to SCCs) is a DAG, a topological order over SCCs always exists. We say that SCC $C_i$ can appear before SCC $C_j$ in such an order iff there is no directed path from $C_j$ to $C_i$ in the condensation graph. Please note that the topological ordering is generally not unique; for example, for the condenstation graph $C_3\leftarrow C_1\rightarrow C_2$, both orderings $(C_1,C_2,C_3)$ and $(C_1,C_3,C_2)$ are valid. We added graph definitions in Appendix A.1.
>
> $\mathbf{x}_{\infty}$ denotes a random vector distributed according to the stationary distribution of the OU process.
>
> The operator $\oplus$ denotes the concatenation of subvectors corresponding to disjoint index sets.
>
> > (10) assumes that we have access to observational and interventional moments. How sensitive is this model to the measurement errors?...
>
> In Eq. (10), we use empirical steady-state moments. We evaluated sensitivity by varying the sample size from 5000 to 40000 and added the results in Appendix B.3; We observe that performance improves substantially when increasing the number of samples (observational data) from 5k to 20k, and then stabilizes. This is expected, since our estimator relies on
> empirical means and covariances of the steady state. With few samples (e.g., 5k), both $\hat{\mu}$ and $\hat{\Sigma}$ are still noisy, leading to moderate parameter errors, whereas for 10k–20k samples, the estimated moments become accurate enough that additional data yield only diminishing returns. In this regime, the remaining error might be due to the fact that we solve the least-squares problem numerically and may not reach the exact optimizer.
>
> A more systematic treatment of moment uncertainty could be obtained by a distributionally robust formulation, in which the loss in Eq. (10) is optimized in a worst-case sense over small uncertainty sets around the empirical means and covariances. We view such an extension as an interesting direction for future work.

---

> ### Author Response · Authors · 2025-11-20
> **Response - Part 2**
>
> > In experiment on a real dataset states that the scaling factor cancels out when estimating the first-order moment
> . But would it also cancel when estimating the covariance?...
>
> Our identifiability result says that any alternative parameter triple consistent with the
> observational and interventional steady state is of the form
> $\hat{\Lambda} = c \Lambda,\quad \hat{b} = c b,\quad \hat{D} = c D,$
> for some scalar $c>0$. For the steady-state mean, we have $\mu = \Lambda^{-1} b, \quad \hat{\mu} = \hat{\Lambda}^{-1} \hat{b} = (c\Lambda)^{-1} (c b) = \Lambda^{-1} b = \mu,$ so the scaling cancels out, as noted in the paper.
>
> The same holds for the steady-state covariance. $\Sigma$ is defined as the unique solution of the
> Lyapunov equation $\Lambda \Sigma + \Sigma \Lambda^\top = D,$ under the assumption that $\Lambda$ is positive stable. Under the scaled parameters, the covariance $\hat{\Sigma}$ satisfies $\hat{\Lambda} \hat{\Sigma} + \hat{\Sigma} \hat{\Lambda}^\top
> = \hat{D}\Longleftrightarrow
> c\Lambda \hat{\Sigma} + \hat{\Sigma} c\Lambda^\top = cD
> \Longleftrightarrow \Lambda \hat{\Sigma} + \hat{\Sigma} \Lambda^\top = D.$
> By the uniqueness of the solution to the Lyapunov equation for a stable drift matrix, we obtain $\hat{\Sigma} = \Sigma$. The same argument applies to any intervened drift $\tilde{\Lambda}^{(i)}$, so the predicted
> steady-state covariances for unseen interventions are also invariant to the global scaling.
>
> > Which assumptions from Thm. 3 can be relaxed without losing identifiability?
>
> The structural conditions in Theorem 3, namely (i) the condensation graph over SCCs being connected and (ii) having a single root SCC, are necessary for identifiability, as explained in Remark 3.
>
> Regarding the spectral non-degeneracy assumptions (e.g., simple spectrum of the drift matrix or of certain moment-based matrices), we believe these assumptions hold generically within each SCC. While we do not provide a formal proof of genericity, our numerical experiments (in Appendix C) indicate that these conditions are satisfied with measure one for randomly generated SCCs. Therefore, these assumptions could in principle be relaxed once a formal genericity argument is established.

---

### Meta-Review · Area_Chair_6qGb · 2026-01-06

**Summary:**

This paper studies identifiability and estimation of multivariate linear OU dynamics from steady-state observational and hard-interventional snapshots, with the headline claim that one intervention per SCC suffices to identify the system up to a global scale.

**Reviewer Concerns:**

Reviewers generally find the theoretical framing clean and potentially useful for experimental design, and agree the core result is simple and interpretable; however, multiple reviewers are borderline due to (i) strong realism gaps: perfect “row-zeroing” interventions, diagonal diffusion/no latent confounding, linearity; (ii) limited empirical breadth and baselines: experiments feel more illustrative than comparative, with insufficient benchmarking against classical non-interventional or alternative moment-based estimators; and (iii) presentation/accessibility issues: missing key definitions in the main text, heavy proof density, and assumptions hard to map to theorems. The rebuttal addresses several technical questions well—especially clarifying that the global scaling does not affect predicted steady-state means/covariances and adding some additional empirical evidence on intervention counts—but it does not fully resolve the main concerns about robustness to imperfect interventions/correlated noise or the lack of broader comparisons.

**Reviewer Scores:**

Reviewer WHmi: may stay at 6.
The rebuttal directly answered the scaling question and gave additional evidence on SCC/intervention counts, which likely removes some uncertainty; however, WHmi’s biggest weaknesses were insufficient baselines and more systematic empirical validation of the “one per SCC” guideline.

Reviewer zTTa: likely 4 -> 6.
This reviewer explicitly said they would increase the score after the rebuttal, and the authors addressed most of their questions, including related work positioning, SCC definition, how to use results for experimental design when SCCs are unknown, and limitations of perfect interventions.

Reviewer iMRJ: may stay at 4.
Remaining issues include diagonal diffusion realism/correlated noise, main-text accessibility rather than appendix-only fixes.

Reviewer MWjF: may stay at 4.
The rebuttal acknowledges limitations and defends scope, but does not add the kind of new comparative evidence.

---

### Decision · Program_Chairs · 2026-01-26

Reject